# Promoting water dissociation for efficient solar driven CO$_2$ electroreduction via improving hydroxyl adsorption

Xin Chen[1], Junxiang Chen[2], Huayu Chen[3], Qiqi Zhang [1], Jiaxuan Li[1], Jiwei Cui[1], Yanhui Sun[1], Defa Wang[1], Jinhua Ye [1,4] & Lequan Liu [1] ✉

Exploring efficient electrocatalysts with fundamental understanding of the reaction mechanism is imperative in CO$_2$ electroreduction. However, the impact of sluggish water dissociation as proton source and the surface species in reaction are still unclear. Herein, we report a strategy of promoting protonation in CO$_2$ electroreduction by implementing oxygen vacancy engineering on Bi$_2$O$_2$CO$_3$ over which high Faradaic efficiency of formate (above 90%) and large partial current density (162 mA cm$^{-2}$) are achieved. Systematic study reveals that the production rate of formate is mainly hampered by water dissociation, while the introduction of oxygen vacancy accelerates water dissociation kinetics by strengthening hydroxyl adsorption and reduces the energetic span of CO$_2$ electroreduction. Moreover, CO$_3$* involved in formate formation as the key surface species is clearly identified by electron spin resonance measurements and designed in situ Raman spectroscopy study combined with isotopic labelling. Coupled with photovoltaic device, the solar to formate energy conversion efficiency reaches as high as 13.3%.

Electrochemical CO$_2$ reduction (CO$_2$RR) to valued chemicals coupled with renewable energy power generation is considered as a promising and elegant step in carbon capture, utilization and storage (CCUS) technologies[1,2]. Among CO$_2$RR products, formate with high energy density is regarded as the most cost-effective and commercially profitable product based on technoeconomic analysis, which has been attracting great attentions[3–5]. In pursuit of the commercial goal, developing efficient catalysts with high selectivity for CO$_2$RR to formate is prerequisite. Though great endeavors have been undertaken, the catalytic performance is still unsatisfactory[6,7]. Thus, developing robust strategies to design efficient electrocatalysts and understanding the reaction route are imperative tasks.

The process of CO$_2$ electroreduction to hydrocarbons involves multistep proton-coupled electron transfer process and a series of adsorbed species. Water as proton source has been clarified while the sluggish water dissociation which will seriously hinder the overall reaction rates of CO$_2$RR is commonly overlooked so far[8–11]. Moreover, the local alkaline condition in CO$_2$RR caused by proton depletion and slow H$^+$ diffusion will further elevate the barrier of H$_2$O dissociation[12,13]. From this perspective, promoting water dissociation through rational material design is attractive and promising to achieve efficient CO$_2$RR, which is preliminarily supported by recent studies[14–16]. Nevertheless, exploring intrinsic active sites of electrocatalysts instead of extra additions for promoting water dissociation and understanding the

[1]TJU-NIMS International Collaboration Laboratory, School of Materials Science and Engineering, Key Lab of Advanced Ceramics and Machining Technology (Ministry of Education), Tianjin University, Tianjin, P. R. China. [2]CAS Key Laboratory of Design and Assembly of Functional Nanostructures, Fujian Key Laboratory of Nanomaterials, Fujian Institute of Research on the Structure of Matter, Chinese Academy of Sciences, Fuzhou, P. R. China. [3]College of Materials and Chemistry, China Jiliang University, Hangzhou, P. R. China. [4]International Center for Materials Nanoarchitectonics (WPI-MANA), National Institute for Materials Science (NIMS), 1-1 Namiki Tsukuba, Japan. This study is dedicated to Professor Jinhua Ye on the occasion of her 60th birthday.
✉e-mail: Lequan.Liu@tju.edu.cn

mechanism of water dissociation in $CO_2RR$ further are highly challenging but desirable.

For electrode materials, metal oxide attracts broad attentions due to high selectivity and low overpotentials which are the crucial parameters for commercial scale in $CO_2RR$ to formate[17–23]. Moreover, recent studies demonstrate that surface metal oxide over metal electrocatalysts largely promotes the catalytic performance in $CO_2RR$, which highlights the critical role of metal oxide in developing efficient catalyst and excites the exploration of mechanism behind[24,25]. As a universal intrinsic defect, oxygen vacancy ($V_O$) in metal oxide is commonly considered to modify metal active site, while the role of $V_O$ itself is neglected generally[26–31]. Yet $V_O$ possesses strong oxygen affinity and fast interaction with water, which indicates it can serve as promising active site in tuning the energy barrier of water dissociation[32,33]. Thus, it is anticipated that constructing $V_O$ might enable enhancing $CO_2$ performance by relieving the effect of sluggish $H_2O$ dissociation and optimizing active sites, which is of great significance in offering a new avenue in efficient electrocatalyst exploration for $CO_2RR$ and deeper understanding the role of water dissociation in the whole reaction. Another main obstacle in the investigation on complex $CO_2RR$ reaction route is the identification of initial surface species. Previous studies propose that $CO_2$ tends to adsorb on oxygen site in metal oxide to form $CO_3$ and participates in sequent reduction process as the key surface species, which is considered to be the origin of attractive performance for metal oxide electrocatalysts[34–38]. However, compelling evidence on the participation of $CO_3$ is still in absence due to the complicated environment in $CO_2RR$ and ambiguous identification of species.

Therefore, $Bi_2O_2CO_3$ (BOC) is selected here based on the following two considerations: (i) there are abundant Bi-O bonds for implementing $V_O$ engineering to promote water dissociation in $CO_2RR$; (ii) the natural carbonate species in BOC is favorable to clarify whether $CO_3$ is involved in $CO_2RR$ with the help of isotopic labeling. Through introducing $V_O$, the production rate of formate reaches 3.0 mmol h$^{-1}$ and the high Faradaic efficiency of formate keeps well over a wide potential window. Dynamic study and DFT calculations reveal the crucial role of water dissociation in promoting $CO_2RR$ kinetics while the introduction of $V_O$ expedites the water dissociation kinetics through improving OH* adsorption which notably reduces the energetic span of formate formation. Theoretical analysis shows $CO_3$* participates in the formation of formate as the key surface species, which is demonstrated clearly through electron spin resonance (EPR) measurements and in situ Raman spectroscopy study combined with isotopic labelling. Finally, full-cell electrocatalysis coupled with solar cell was constructed and achieves the solar to formate energy conversion efficiency of 13.3%.

## Results

### Material synthesis and the identification of $V_O$

The $Bi_2O_2CO_3$ samples supported on carbon paper were prepared by electrodeposition method. $V_O$ enginnering was achieved by tuning the proportion of water in the mixed electrolyte with ethylene glycol, and a series of samples denoted as BOC-1, BOC-2, BOC-3 and BOC-4 respectively, were obatined[39]. The tetragonal structure of $Bi_2O_2CO_3$ is clearly identified from X-ray diffraction (XRD) patterns for the synthesized samples (JCDPS 41-1488, Supplementary Fig. 1). As revealed by scanning electron microscopy (SEM), the BOC samples are grown on carbon paper vertically, exhibiting stacked and curved nanosheet morphology (Fig. 1a and Supplementary Fig. 2). Transmission electron microscope (TEM) images show that the average lateral size of BOC nanosheets ranges from 100 to 200 nm (Fig. 1b and Supplementary Fig. 3). A lattice distance of 0.275 nm corresponding to (110) plane is clearly discerned in high-resolution TEM (HRTEM) images (Fig. 1c), while the selected area electron diffraction (SAED) patterns display that the BOC nanosheets are consisted of individual single crystalline

nanosheet (inset in Fig. 1c and Supplementary Fig. 3d). Notably, variation and distortion in lattice fringes are preliminarily observed, indicating the presence of defects which might be $V_O$ (Supplementary Fig. 4).

Aberration-corrected scanning transmission electron microscopy (STEM) was conducted to disclose the fine structure and verify the defects. Local lattice disorders in nanosheets are clearly detected in high-angle annular dark-field (HAADF) images (Fig. 1d, e and Supplementary Fig. 5), owing to the unsaturated coordination of metal atoms[40,41]. With the help of angular bright-field STEM (ABF-STEM), the nonperiodic intensity of oxygen can be discerned, demonstrating the presence of $V_O$ as marked in Fig. 1f, g and Supplementary Fig. 6. The regular variation of the sharp signal intensity with g value of 2.002 in EPR characterization not only further clarifies the presence of $V_O$ but also indicates that $V_O$ concentration increases in the sequence of BOC-1, BOC-2, BOC-3 and BOC-4 (Fig. 1h)[42]. Commercial BOC (denoted as BOC-C) was adopted for comparison, and the $V_O$ concentration is much lower than that of BOC-1 as suggested by the weak EPR signal. The $V_O$ concentration was also quantified from EPR while atomic $V_O$ contents are estimated to be 0.031%, 0.044%, 0.060% and 0.076% for BOC-1, BOC-2, BOC-3 and BOC-4, respectively (Supplementary Table 1)[43]. The location of $V_O$ is disclosed from extended X-ray absorption fine structure (EXAFS) spectroscopy, and two peaks at around 1.6 Å and 3.5 Å corresponding to the scattering path of Bi-O and Bi-O-C are found, respectively (Fig. 1i, Supplementary Fig. 7 and Supplementary Table 2). The intensity of Bi-O manifests a decrease for BOC with more $V_O$ contents while there is no obvious difference in that of Bi-O-C, which implies that $V_O$ mainly exists in Bi-O-Bi structure[44]. X-ray absorption near edge structure (XANES) region (Fig. 1j) shows that the absorption edge for BOC with $V_O$ just slightly shifts to lower energy, which is in agreement with XPS spectra of Bi $4f$ and indicates the presence of low chemical state Bi induced by $V_O$ (Fig. 1k)[45]. Based on the characterizations above, it can be concluded that the BOC with different contents of $V_O$ is successfully prepared.

### The evaluation of $CO_2RR$ performance

Electrocatalytic $CO_2$ reduction was first evaluated in the traditional H-type cell using 0.5 M $KHCO_3$ as electrolyte. The linear sweep voltammetry (LSV) curves of all samples in $CO_2$-saturated electrolyte show that BOC-2 exhibits superior $CO_2RR$ performance with smaller potentials and larger current density (Fig. 2a). Then, the selectivity and activity as function of potentials were testified via chronoamperometry (Supplementary Fig. 8). For BOC-C, the Faradaic efficiency of formate ($FE_{formate}$) is 7.6% (Fig. 2b) with the partial current density of formate ($j_{formate}$) of 0.04 mA cm$^{-2}$ at −0.68 V vs. RHE (unless mentioned specifically, all potentials referred are versus reversible hydrogen electrode hereafter), which is consistent with early report[28]. With the introduction of $V_O$, the $FE_{formate}$ increases remarkably to 90.2% at −0.68 V while the $j_{formate}$ raises to 16.5 mA cm$^{-2}$ with the increase of $V_O$ concentration up to 0.044 at.%, which demonstrates that the introduction of $V_O$ can enhance the selectivity and activity of $CO_2RR$ to formate. At −1.08 V, the maximum $j_{formate}$ of BOC with $V_O$ is three times as high as that of BOC-C (Fig. 2c). The further increase of $V_O$ concentration makes the $FE_{formate}$ and production rate of formate drop accompanied with notable $FE_{hydrogen}$ (Fig. 2d and Supplementary Fig. 9). Nevertheless, the performance of BOC samples with excessive $V_O$ contents can be elevated through filling $V_O$ partially, which was achieved through thermal treatment in air at 200 °C (these annealed samples are denoted as BOC-A) with the maintenance of phase structures (Supplementary Fig. 10). As compared with original BOC, the decrease of $V_O$ concentration in all BOC-A samples is verified by EPR measurement and the signal of BOC-1-A is hardly to be discerned, indicating the $V_O$ is almost eliminated in BOC-1-A as shown in Supplementary Fig. 11. For $CO_2RR$ evaluated in the same set-up (Supplementary Fig. 12 and Supplementary Fig. 13), both the $FE_{formate}$ and

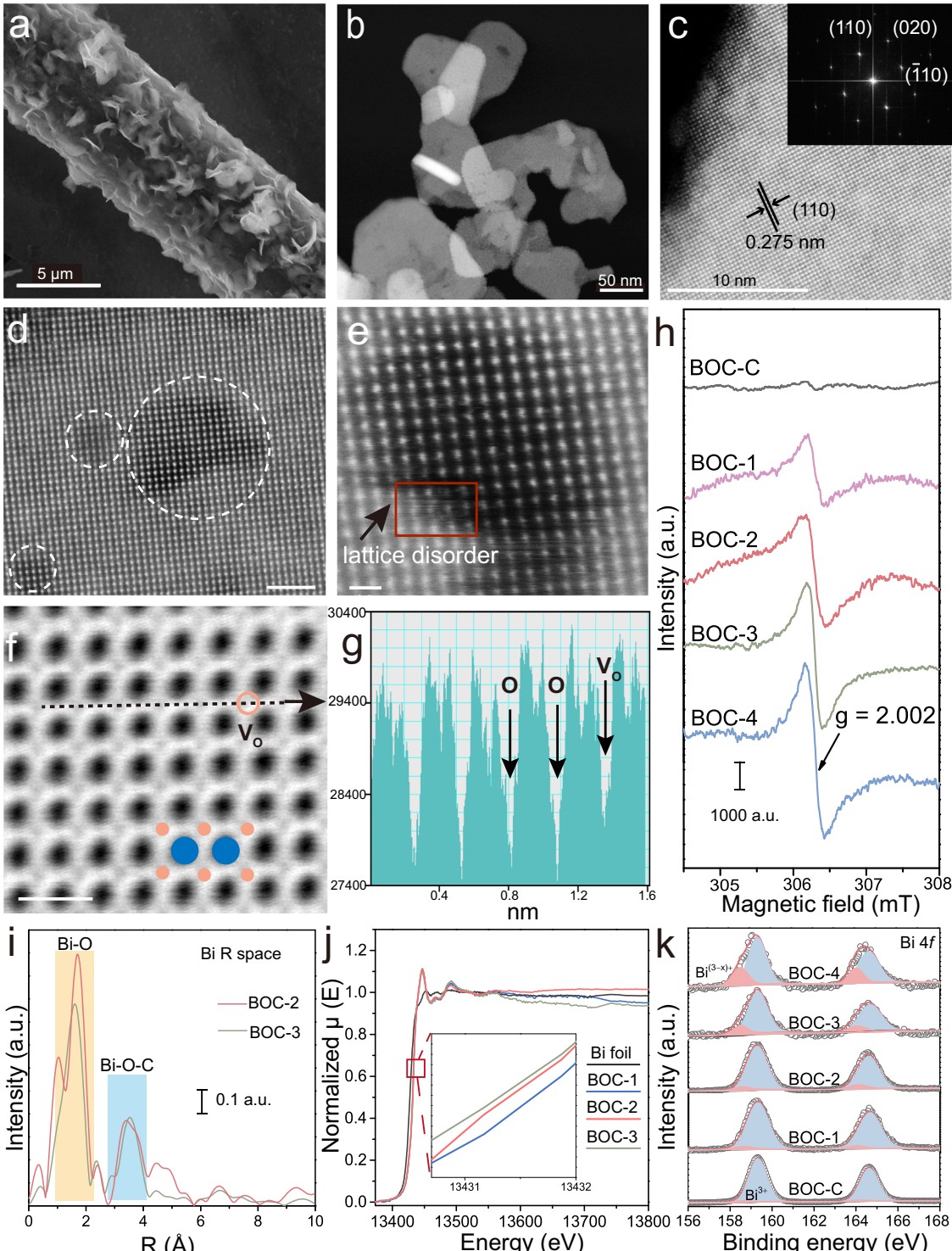

**Fig. 1 | Characterizations of BOC and oxygen vacancy. a** SEM, (**b**) TEM and (**c**) HRTEM images of BOC-2. **d**, **e** Atomic-resolution HAADF-STEM images of BOC-2, where the circled areas show the lattice disorders. The scale bar is 2 nm in (**d**) and 0.5 nm in (**e**), respectively. **f** ABF-STEM image of BOC-2, where the pink and blue circles represent oxygen and bismuth atoms, respectively. The scale bar is 0.5 nm. **g** Intensity profile corresponding to the black dashed line in (**f**), as directed by the arrow. **h** EPR spectra of all samples, where the g value of 2.002 is the characteristic signal of $V_O$. **i** Fourier transform of Bi $L_3$ edge EXAFS data recorded at R space. **j** Normalized XANES spectra of Bi foil, BOC-1, BOC-2 and BOC-3. Inset: the magnified area marked by red line. **k** XPS spectra of BOC-C, BOC-1, BOC-2, BOC-3 and BOC-4 on Bi 4$f$, where the blue and pink area represent $Bi^{3+}$ and $Bi^{(3-x)+}$, respectively.

current density are promoted for BOC-3-A and BOC-4-A with reduced $V_O$ concentration (Fig. 2e and Supplementary Fig. 14). In particular, the production rate of formate for BOC-4-A is 2.8 times higher than that of initial BOC-4 without annealing (Fig. 2d, black line). These experiments establish an obvious correlation between $V_O$ and formate production,

which proves that promoting $CO_2RR$ to formate over BOC can be achieved through $V_O$ engineering.

BOC-2 with the superior performance of $CO_2RR$ is studied further at different potentials. The $FE_{formate}$ is 70% at −0.58 V initially and increases rapidly to 94% with current density of 34.6 mA cm$^{-2}$ at

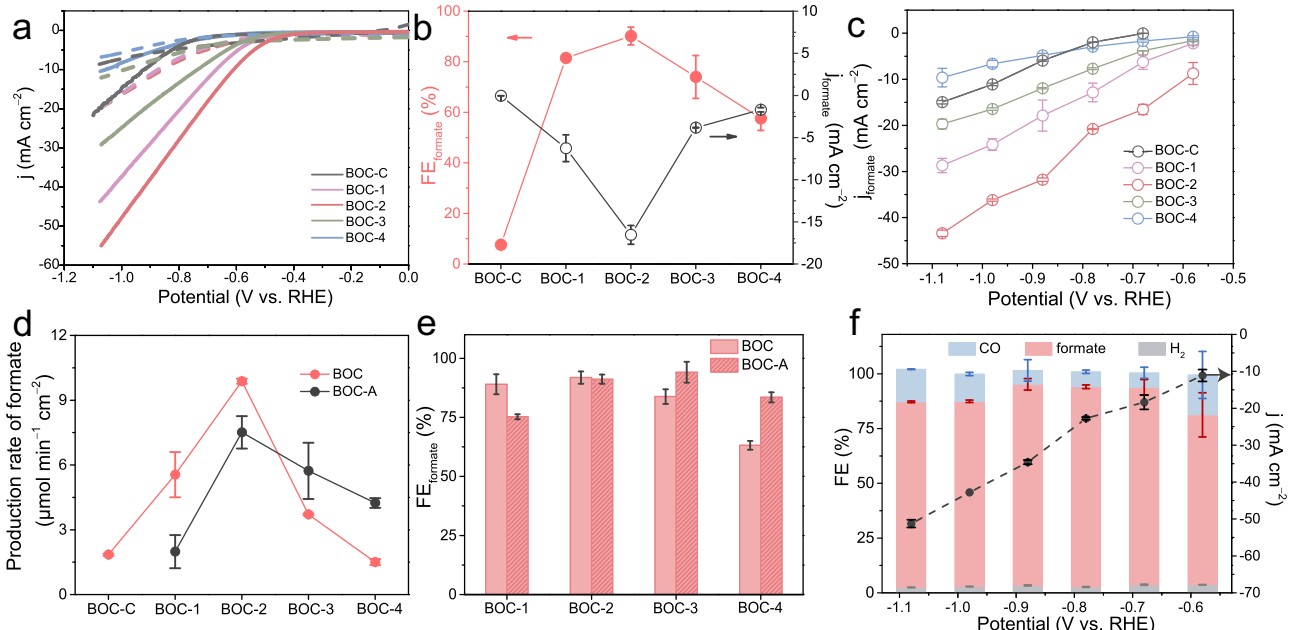

**Fig. 2 | Electrochemical CO₂RR performance. a** LSV curves of BOC-C, BOC-1, BOC-2, BOC-3 and BOC-4 in Ar-saturated (dashed lines) and CO₂-saturated (solid lines) 0.5 M KHCO₃ solutions. **b** FE$_{formate}$ and j$_{formate}$ of BOC with differnt V$_O$ concentrations at −0.68 V. **c** j$_{formate}$ on different samples as function of applied potentials. **d** The production rate of formate for all samples at −0.88 V. The red and black lines represent samples before and after thermal treatment, respectively. **e** FE$_{formate}$ of all samples before and after thermal treament at −0.88 V. **f** The FE of all products and current density for BOC-2 at different applied potentials in H-cell. The error bars represent the standard deviation of three independent experiments.

−0.88 V while HER is almost negligible, outperforming most of reported Bi-based catalysts (Fig. 2f and Supplementary Table 3). Moreover, the FE$_{formate}$ for BOC-2 is above 80% at a wide potential window (−0.68 V to −1.08 V) despite slight decay at more negative potentials due to the restriction of CO₂ solubility and mass transfer. During 11 h operation in H-cell, the FE$_{formate}$ above 90% for BOC-2 could be maintained with formate yield reaching 6.5 mmol (Supplementary Fig. 15). Meanwhile, there is no obvious change in phase structure, morphology and valance state, which demonstrates the stability of BOC-2 (Supplementary Fig. 16). To relieve the limitation on mass transfer and pursue commercial current density, BOC-2 was further integrated into gas diffusion electrode (GDE) and evaluated in flow-cell system[46]. The j$_{formate}$ of BOC-2 is 1.6 times as high as that of BOC-C at −1.68 V while the turnover frequency can be up to 0.72 s$^{-1}$ at 200 mA cm$^{-2}$, which demonstrates the activity enhancement through introducing V$_O$ can also be achieved in flow-cell system (Supplementary Fig. 17 and Supplementary Table 4). In addition, a stable FE$_{formate}$ of 80% at large current density of 200 mA cm$^{-2}$ can be observed during 15 h, which indicates the potential for practical application (Supplementary Fig. 18). Even though Bi$^{3+}$ could be reduced at negative potentials in thermodynamics, the good stability of BOC can be explained by the spontaneous CO₂ adsorption and high local pH where the oxide state of Bi is stable[24,28,47,48]. Moreover, $^{13}$CO₂ labeling experiment was carried out. The proton doublet resulting from H-$^{13}$C coupling and H$^{13}$COO$^-$ is observed in $^1$H NMR and $^{13}$C NMR, respectively (Supplementary Fig. 19), demonstrating that the produced formate derives from CO₂[49]. Trace amount of formate produced in Ar-saturated KHCO₃ should be attributed to slight CO₂ decomposed by HCO₃$^-$ in electrolyte (Supplementary Fig. 20). These results reveal that, through V$_O$ engineering, BOC-2 demonstrates efficient CO₂RR to formate with high selectivity, large current density and stability.

## Investigation about the effect of V$_O$ in CO₂RR

To clarify the intrinsic activity of BOC with V$_O$, the j$_{formate}$ of different samples were normalized by specific surface area

(Supplementary Figs. 21, 22) and electrochemical surface area, respectively (Supplementary Fig. 23 and Supplementary Table 5). It can be found that the introduction of V$_O$ notably increases the intrinsic activity and the degree of this enhancement is closely related with the V$_O$ contents (Supplementary Fig. 24). Tafel plots were then obtained at sufficient low overpotential to investigate the role of V$_O$ in the kinetics of CO₂RR. As can be seen from Fig. 3a, BOC-2 shows smaller Tafel slope among all catalysts, indicating that the introduction of V$_O$ favors the kinetics of CO₂RR[50]. It's noteworthy that the value of Tafel slope is much larger than the reported typical value, which indicates that the chemical step precedes the electron transfer or the chemical step is rate-limiting. The detailed investigation of the reaction mechanism will be discussed later. Next, electrochemical impendence spectroscopy (EIS) was carried out and fitted by equivalent circuit to investigate the electrochemical interface properties[51,52]. With the introduction of V$_O$, the charge transfer can be accelerated, which is beneficial for conductivity and reducing the overpotential (Supplementary Fig. 25, Supplementary Table 6). Besides, the resistance induced by water and hydroxyl adsorption (R$_p$) is related with V$_O$ concentration, implying that V$_O$ might affect the proton transfer by changing water and hydroxyl adsorption. Before conducting the investigation on proton transfer further, the proton source was clarified through isotopic labelling of D₂O. The product in KHCO₃-D₂O is almost DCOO$^-$ while the signal of HCOO$^-$ in $^1$H NMR is negligible, which demonstrates that the dominate source of proton is from water instead of HCO₃$^-$ (Supplementary Fig. 26). Subsequently, kinetic isotopic effect (KIE) experiments were carried out by varying H₂O and D₂O in electrolyte to study the effect of water dissociation and proton transfer in CO₂RR (Fig. 3b and Supplementary Fig. 27). The KIE value is calculated by the ratio of formate production rate in KHCO₃-H₂O and KHCO₃-D₂O. In general, KIE value is >1 if water dissociation is involved in rate-limiting process of formate production due to the proton tunneling effect, while the higher KIE value indicates the greater impact of water dissociation in CO₂RR (Supplementary Fig. 28 and detailed discussion can be seen in Method section)[53]. For

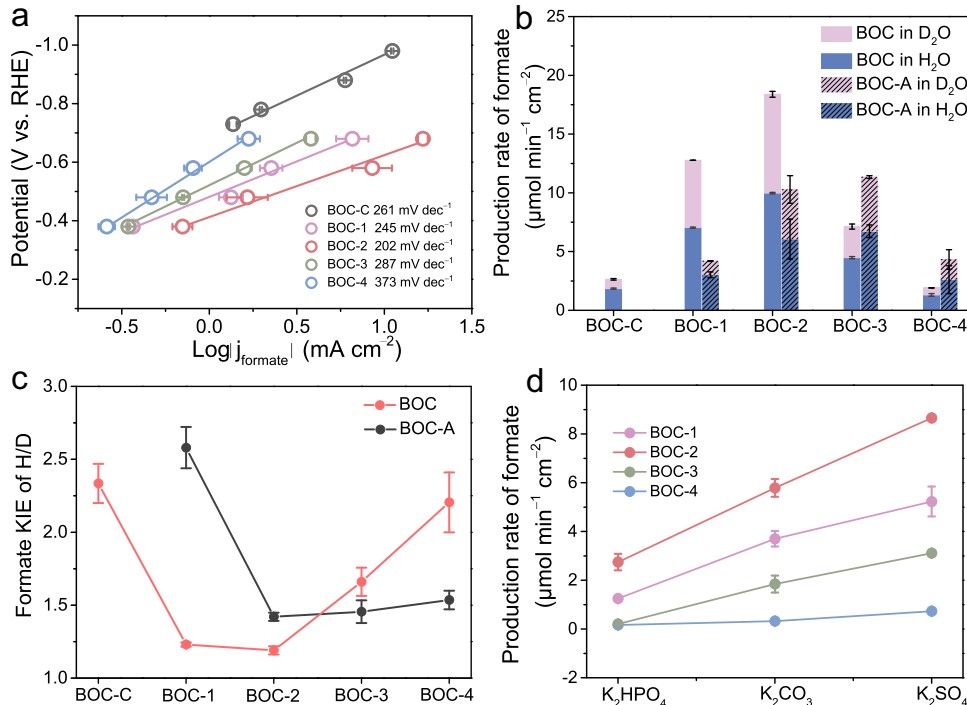

**Fig. 3 | Mechanism analysis. a** Tafel plots of different samples. **b** The production rates of formate in 0.5 M KHCO₃-H₂O (blue column) and 0.5 M KHCO₃-D₂O (purple column). The columns with shadows represent samples after thermal treatment. **c** The KIE value of H/D for different samples. The red and black lines represent

BOC-C, the KIE value was determined to be 2.34, suggesting the reaction rate of $CO_2RR$ is predominantly limited by water dissociation (Fig. 3c). With the introduction of $V_O$, the KIE value of BOC-2 drops rapidly to 1.19, which reveals that $V_O$ can remarkably promote water dissociation and subsequently enhance $CO_2RR$ activity. Nevertheless, the KIE value of BOC-3 and BOC-4 with higher $V_O$ concentration increase again, indicating that excessive $V_O$ is unbeneficial to water activation. Interestingly, the ability of water dissociation for these two samples could be elevated by partially filling $V_O$ (Supplementary Fig. 29), which can be seen from the lower KIE value and enhanced activity in $CO_2RR$ (see BOC-3-A and BOC-4-A). These results agree well with the formate production rates dissused above and a strong correlation among $V_O$, water dissociation ability and formate production is identified. Besides, less enhancement in activity induced by cations can be observed on BOC-2 as compared with that on BOC-C, indicating the intrinsic faster water dissociation kinetics of BOC-2 (Supplementary Fig. 30)[54]. The role of $V_O$ in water dissociation was further clarified through varying local pH achieved by using three electrolytes with different buffer capacity ($K_2HPO_4$, $K_2CO_3$ and $K_2SO_4$). It has been demonstrated that the local pH value increased in the order of $K_2HPO_4$, $K_2CO_3$ and $K_2SO_4$, while water dissociation becomes more difficult at high pH value[15]. The same concentrations of cations among those three kinds of electrolyte can exlude the effect of cations in $CO_2RR$[55,56]. As can be seen from Fig. 3d and Supplementary Fig. 31, the production rate of formate for BOC-2 is much higher than that of other samples at high local pH values, which indicates the beneficial effect of $V_O$. Moreover, it can be found that BOC-2 shows superior activity and selectivity than BOC-C in flow-cell at large current density where water dissociation is more difficult due to the high local pH induced by rapid protons depletion (Supplementary Fig. 17c, d). So, it can be concluded that water dissociation is involved in the rate-determining step for $CO_2RR$ to formate, and the presence of $V_O$ remarkably boosts the performance of $CO_2RR$ by accelerating $H_2O$ dissociation.

samples before and after annealing, respectively. **d** The production rates of formate over BOC-1, BOC-2, BOC-3 and BOC-4 in $K_2HPO_4$, $K_2CO_3$ and $K_2SO_4$ electrolytes. The error bars represent the standard deviation of three independent experiments.

## The investigation of reaction pathway and the role of water dissociation in $CO_2RR$

Density functional theory (DFT) simulation was carried out to gain an deeper understanding of the reaction route and the promoting effect of $V_O$. Computational hydrogen electrode (CHE) method was used to get the reaction free energy diagram (FED) of $CO_2RR$[57,58], while the " virtual energetic span" (denoted as $\delta E^v$) was introduced as the activity determining term[59]. Subsequent to $CO_2$ adsorption ($CO_2 + O^* \rightarrow CO_3^*$, step I), the formation of OCHO* is a relatively complex process at least involving electron transfer, C-O cracking and protonation. More importantly, as demonstrated above, water dissociation plays a crucial role in protonation. So, it is reasonable and necessary to divide the process of OCHO* formation into elemental steps involving water dissociation as follows:

$$H_2O\ (aq) + * \rightarrow H_2O^* \qquad \text{step II}$$
$$CO_3^* + H_2O^* \rightarrow OCHO\cdot OH^* + O^* \qquad \text{step III}$$
$$OH^* + e \rightarrow OH^- \qquad \text{step IV}$$

Take consideration of the two-dimensional property of BOC, both the edge and terrace sites are first investigated, and the FEDs along the whole process of $CO_2RR$ to formate are plotted in Fig. 4a and Supplementary Fig. 32. The $\delta E^v$ of edge sites is much lower than that of terrace sites (0.96 eV vs. 2.35 eV), which indicates that $CO_2RR$ proceeds preferentially at edge sites. Meanwhile, it can be noted that the Step III is the main uphill barrier for BOC, which involves H-OH cleavage in water dissociation ($H_2O^* \rightarrow OH^*$), C-O cracking and C-H formation (Fig. 4b, c). With the introduction of $V_O$ (BOC-$V_O$-1, Fig. 4d), the energy barrrier of step III ($E_{III}$) reduces siginificantly compared with that of normal BOC (0.70 eV vs. 1.66 eV). Correspondingly, the BOC with $V_O$ possesses much smaller $\delta E^v$ (0.61 eV) relative to normal BOC (0.96 eV), which indicates that the introduction of $V_O$ promotes $CO_2RR$ by reducing energetic span (Supplementary Fig. 33). In order to figure out the rate-limiting process in step III, the chemical potential difference between $H_2O^*$ and $OH^*$ (denoted as $\Delta\mu_{OH-w}$), reflecting water dissociation ability, is obtained and plotted with $E_{III}$. As can be deduced

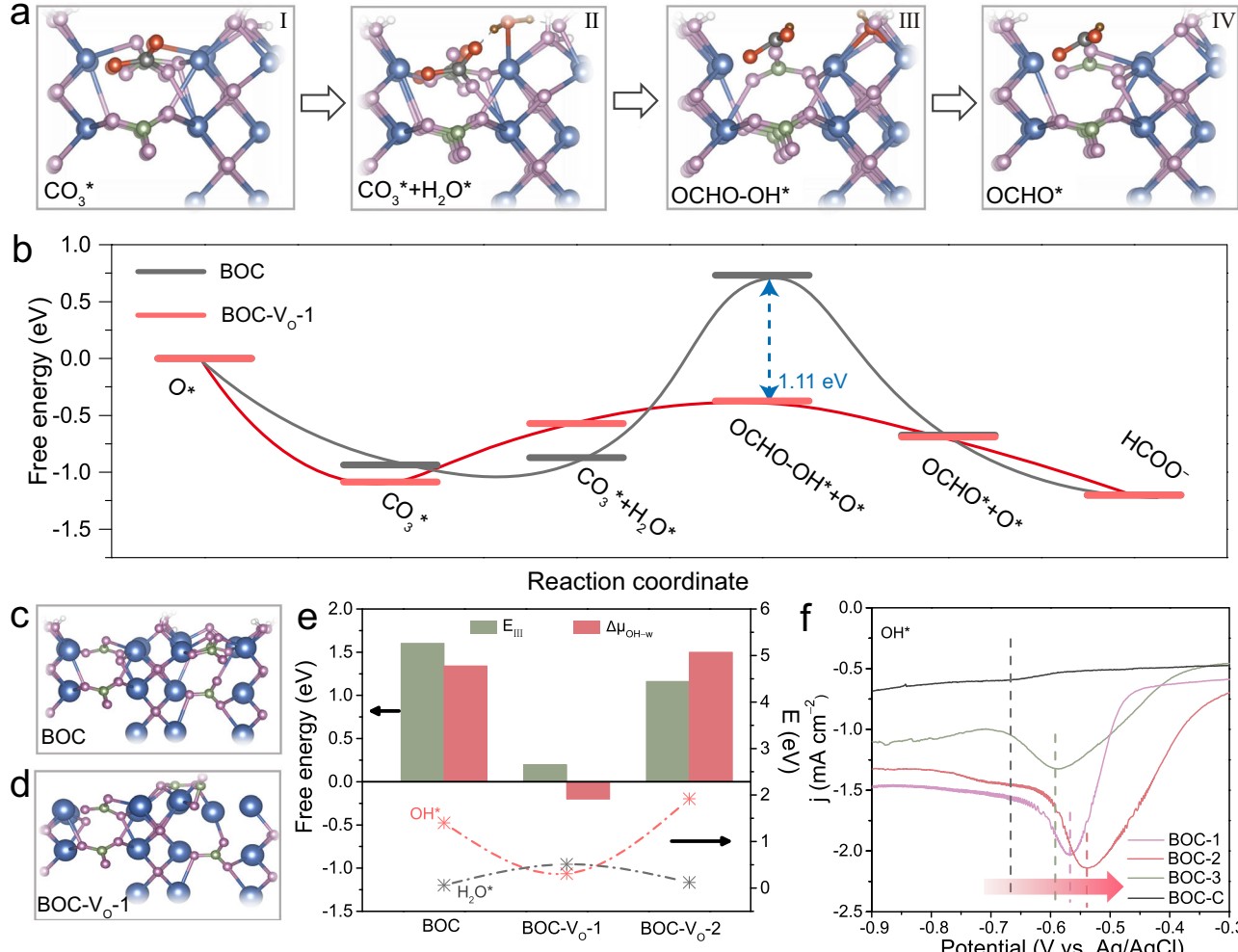

**Fig. 4 | DFT calculations of the reaction pathway and the role of oxygen vacancy in water dissociation. a** The key intermediates during $CO_2RR$ to formate. Colours in the models: blue balls are bismuth (Bi); purple balls are oxygen (O); green balls are carbon (C); white balls are hydrogen (H); red balls are added oxygen ($O_{add}$); grey balls are added carbon ($C_{add}$); brown balls are added hydrogen ($H_{add}$). **b** The free energy plots for BOC with and without $V_O$ along the reaction pathway. **c, d** The basic computed models of normal BOC and BOC with $V_O$ on surface (BOC-$V_O$–1), respectively. **e** The free energy of step III, $\Delta\mu_{OH-w}$, the adsorption energy of OH* and the adsorption energy of $H_2O$* for BOC with different $V_O$ contents. **f** LSV curves of BOC with different $V_O$ concentrations in Ar-saturated 0.1 M KOH.

from Fig. 4e, the energy variation of the step III is mainly from $\Delta\mu_{OH-w}$, which suggests that the formate production rate is dominantly restricted by the sluggish water dissociation. While the introduction of $V_O$ accelerates the kinetics of water dissociation by notably reducing $\Delta\mu_{OH-w}$, which largely decreases the energy barrier in the formation of OCHO*. These results are in good agreement with KIE experiments discussed above. On the other hand, the accelerated water dissociation kinetics has minor effects on HER due to the poor H* adsorption on BOC (Supplementary Fig. 34).

To clarify the cause of decreased $\Delta\mu_{OH-w}$, the adsorption energy of OH* and $H_2O$* are plotted at the bottom of Fig. 4e, respectively. The introduction of $V_O$ significantly reduces the adsorption energy of OH*, i.e., enhancing OH* adsorption, while the change in the energy of water adsorption is limited. That is, the reduced $\Delta\mu_{OH-w}$ is predominately originated from the strengthened OH* adsorption induced by the introduction of $V_O$, which leads to the decrease of $\Delta\mu_{OH-w}$ and easier $CO_2$ protonation. This might be caused by the speculation that OH* formed in step III can alternatively adsorb on the $V_O$ site while the vacancy is not large enough in volume to accommodate $H_2O$*. Based on BOC-$V_O$–1, $V_O$ was further introduced into the subsurface to increase the $V_O$ contents (Supplementary Fig. 35). Excessive $V_O$ makes the barrier in step III rise again with relatively weak OH* adsorption, which should be due to the structure distortion and lower O

coordinate numbers of Bi induced by more $V_O$. To verify the correlation of OH* adsorption energy and $V_O$ in experiments further, OH* adsorption over BOC with differnet $V_O$ contents were investigated through conducting LSV in Ar-saturated 0.1 M KOH solutions[60]. As can be seen from Fig. 4f, BOC with $V_O$ exhibits more positive peak compared with BOC-C, which implies that the introduction of $V_O$ strengthens the OH* adsorption indeed. These simulation and experimental results reveal that the introduction of $V_O$ remarkably promotes water dissociation by strengthening the adsorption of OH*, which reduces the energetic span of the key intermediate of OCHO* formation and ultimately boosts the efficiency of $CO_2RR$ to formate.

For $CO_2$ adsorption and forming surface species, DFT calculations show that intrinsic $CO_3^{2-}$ in BOC will spontaneously undergo charge rearrangement and be in-situ transformed into $CO_3$* which participates in the sequent formate production with a O* site left (Supplementary Fig. 36). Then, $CO_2$ prefers to adsorb on this oxygen site to form $CO_3$* species again than adsorb on bismuth. The charge distribution of $CO_3$* based on Bader charge analysis indicates the $CO_3$* is in the form of radical with an unpaired electron (Supplementary Fig. 37). To clarify the presence of $CO_3$*, EPR measurements were first performed with the trapping agent of 5,5-dimethyl-1-pyrroline-N-oxide (DMPO). In the absence of $CO_2$, there is just the nonet ascribed to H* radicals (hyperfine splitting constants, $A_N = 1.65$ mT, $A_H = 2.25$ mT) which are

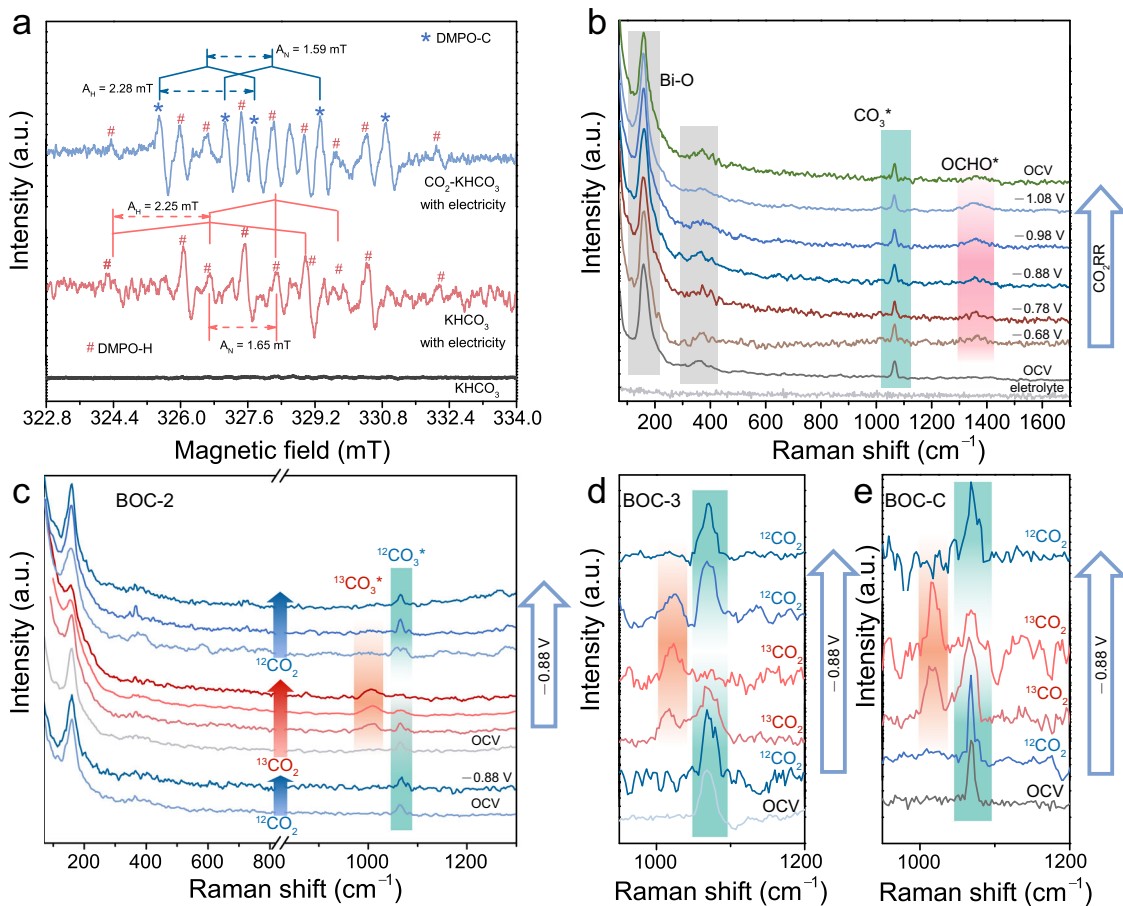

**Fig. 5 | The identification of the participation of CO₃\* in CO₂RR. a** EPR trapping of carbon radicals and hydrogen radicals. **b** In situ Raman spectroscopy for BOC-2 at a series of working potentials. **c**, **d**, **e** In situ Raman spectroscopy with $^{12}CO_2$ and $^{13}CO_2$ as carbon source for BOC-2, BOC-3 and BOC-C, respectively.

originated from water dissociation (Fig. 5a)[61]. With the feed of CO₂, the characteristic sextet of carbon radicals can be found ($A_N$ = 1.59 mT, $A_H$ = 2.28 mT), which agrees well with theoretical analysis[62]. These observations imply the presence of CO₃\* during CO₂RR process.

Then, in-situ Raman measurements were implemented to probe the related species directly along the reaction route. For the intrinsic Raman spectrum of BOC, the bands at 155 cm⁻¹ and 367 cm⁻¹ are the characteristic vibrational modes of Bi=O bond lattice in $[Bi_2O_2]^{2+}$. The peak centered at 1069 cm⁻¹ is attributed to the typical C-O stretching of carbonate in $Bi_2O_2CO_3$ (Supplementary Fig. 38)[63]. At open circuit voltage (OCV), this signal blueshifts to the center of 1067 cm⁻¹, which indicates the transformation of CO₃\* from intrinsic $CO_3^{2-}$ at least occurs under OCV in consideration of DFT calculations and EPR results[64]. Similar results are obtained in the electrolyte of $K_2HPO_4$, which excludes the interference of $HCO_3^-$ and $CO_3^{2-}$ in electrolyte. Then, the Raman spectra of BOC-2 were collected at different potentials (from −0.58 V to −1.08 V) and all these peaks could be kept well without the appearance of metallic Bi signals, demonstrating the stability of BOC-2 under a series of negative potentials (Fig. 5b). Specially, the signal at 1364 cm⁻¹ assigned to the O-C-O symmetric vibration of OCHO\* can be observed at −0.78 V while the intensity of OCHO\* increases continuously with more negative potentials applied, indicating that OCHO\* is the intermediate in formate formation[65,66]. The observation of OCHO\* intermediate provides a strong support for the reaction mechanism proposed in our DFT calculations.

To verify the cycle of CO₃\* in CO₂RR, three steps with isotopic labelling were designed by using $^{12}CO_2$ and $^{13}CO_2$ as the carbon source interchangeably. First, $^{12}CO_2$ was introduced into the reactor, and there is an obvious peak of $^{12}CO_3^*$ accompanied with Bi=O

and $^{12}OCHO^*$ vibration at −0.88 V (Fig. 5c and Supplementary Fig. 39). Subsequent to Ar purging to remove the residual dissolved $^{12}CO_2$ in reactor, $^{13}CO_2$ was introduced into the system for 15 min to achieve saturation. It can be noted that a new and wide peak at 1012 cm⁻¹ appears which indicates the presence of $^{13}CO_3^*$ based on the isotopic effect[67]. The slight difference of peak position between experiments and theoretical calculations may be derived from solvent effect and applied potential[68]. Moreover, the intensity of $^{13}CO_3^*$ increases accompanied with the intensity decrease of $^{12}CO_3^*$ as time goes on, indicating that the initial $^{12}CO_3^*$ in the first step is gradually consumed and replaced by $^{13}CO_3^*$. It's noteworthy that there was no similar exchange at OCV and −0.18 V far away from the onset potential of formate production, which suggests that the observed exchange is indeed resulted from the involvement of CO₃\* in CO₂RR. In the final step, $^{12}CO_2$ was fed into the system again and the intensity of $^{12}CO_3^*$ increases impressively with the gradual disappear of $^{13}CO_3^*$, which manifests clearly that CO₃\* is involved in CO₂RR as the key surface species. To exclude the occasionality of BOC-2, BOC-3 with higher $V_O$ concentration and BOC-C were also tested under the same process (Fig. 5d, e). The experimental results are well consistent with the study discussed above, which indicates that the participation of CO₃\* is universal in CO₂RR. The partial exchange in BOC-C after $^{13}CO_2RR$ indicates that there are some $CO_3^{2-}$ fail to be transformed into the active CO₃\*, which may be one of the reasons for the poor activity of BOC-C. Besides, the signal of CO₃\* disappears when CO₂ was replaced by Ar, which suggests the involvement of CO₃\* as well (Supplementary Fig. 40). Based on the characterization and discussion above, it can be concluded that CO₃\* is involved in formate production as the key surface species.

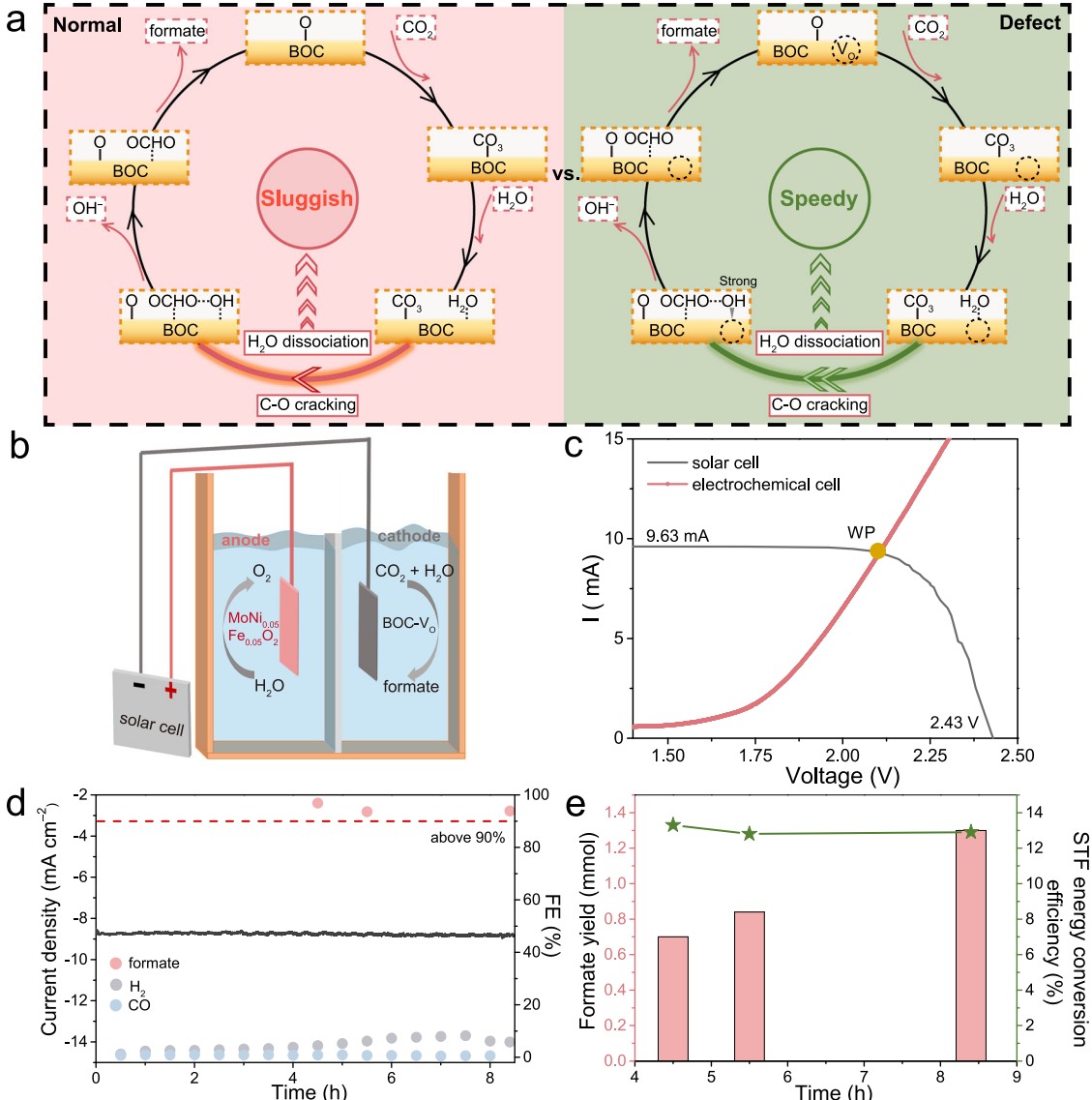

**Fig. 6 | The schmatic mechanism illustration for BOC during CO₂RR to formate and the construction of PV-EC system. a** The schematic plot of the reaction pathways and the role of V_O in CO₂RR. **b** The schematic illustration of PV-CO₂RR system. **c** I−V curves of the solar pannel and the polarization curves of this full-cell electrochemical system. **d** Stability test of PV-CO₂RR system and correponding current density as well as Faradaic efficiency. **e** The formate production and solar to formate energy conversion efficiency during stability test.

In brief, $CO_2$ tends to adsorb on oxygen site to form $CO_3^*$ species which participates in the sequent reduction process where the uphill barrier is mainly originated from the sluggish water dissociation kinetics. The introduction of $V_O$ favors $OH^*$ adsorption and reduces the energy barrier on H-OH cracking, which results in accelerating water dissociation and enhances the formate production rates ultimately. The schematic diagram in Fig. 6a plots the overall reaction routes and the promoting role of $V_O$ in CO₂RR to formate.

## The coupling of photovoltic device and electrochemical CO₂RR (PV-EC system)

In our PV-EC system, the electrochemical CO₂RR was driven by an commercial GaInP/GaAs/Ge solar cell under the illumination of a 100 mW cm$^{-2}$ (1 Sun) solar simulator (Fig. 6b). BOC-2 with the area of $1 \times 1$ cm$^2$ worked as cathode while $MoNi_{0.05}Fe_{0.05}O_2$ was chosen as anode due to the lower overpotentials in oxygen evolution reaction[69]. According to the node of the current-voltage (I-V) curve and the polarization curve of full $CO_2$-$H_2O$ system, the actual working potential (WP) can be estimated to be 2.1 V with the current density of

9.1 mA cm$^{-2}$ (Fig. 6c). Notably, this working outpower is close to the maximum outpower point (MPP) with the solar photovoltaic conversion efficiency of 23.0% (Supplementary Fig. 41). During 8 h operation, this full-cell system with the only energy input of solar can keep stable with the average current density of 8.8 mA cm$^{-2}$, while the $FE_{formate}$ can be above 93% on average (Fig. 6d). The solar to formate (STF) conversion efficiency is as high as 13.3% with the average STF up to 12.9%, outperforming the typical efficiency in solar to formate conversion (Fig. 6e; Supplementary Table 7).

In summary, we demonstrate an effective strategy to promote CO₂RR to formate by accelerating water dissociation through oxygen vacancy engineering. The BOC-2 exhibits $FE_{formate}$ of 94% with partial current density of 32.5 mA cm$^{-2}$. Equipped with flow-cell system, the production rate of formate can reach as high as 3.0 mmol h$^{-1}$ at 200 mA cm$^{-2}$. KIE experiments and DFT calculations reveal that the production of formate is predominantly hindered by sluggish water dissociation serving as proton source. The in-depth theoretical analysis and dynamic experiments as well as electrochemical analysis demonstrate that the introduction of $V_O$ remarkably promotes water

dissociation via strengthening hydroxyl adsorption which reduces the $\Delta\mu_{OH-w}$ and energetic span of formate formation. Moreover, the participation of $CO_3^*$ ($CO_2$ adsorbs on oxygen site) in $CO_2RR$ as the key surface species is clearly confirmed by EPR tests and in situ Raman spectroscopy study with the help of isotopic labeling. Finally, the full-cell electrocatalysis driven by solar cell was constructed with the STF reaching 13.3%. We hope that this overall study of reaction process and effect of water dissociation would motivate more efficient strategies to boost $CO_2RR$ activity in the future.

## Methods

### Synthesis of BOC samples

The BOC with different $V_O$ concentrations were synthesized by electrodeposited method. In the typical synthesis, 0.02 M $BiCl_3$ (dissolved in the mixed solution of ethylene glycol and water) was prepared as electrolyte and cooled in refrigerator at 0 °C for 2 h in advance. Carbon paper was cut into $1 \times 2\ cm^2$ and immersed into the electrolyte with the area of $1 \times 1\ cm^2$ as working electrode. The cathodic current of 40 mA was applied to the carbon paper for 15 mins by galvanostatic method while $CO_2$ was bubbled into the electrolyte during the whole process. The electrode was removed and cleaned with deionized water subsequently. Promoting the breakage of Bi-O bond through varying solvent is in favor of increasing the $V_O$ concentration. Here, by varying the volume ratio of ethylene glycol and water in electrolyte, BOC-1 (pure ethylene glycol), BOC-2 (3:1), BOC-3 (1:1) and BOC-4 (pure water) were obtained, respectively. Before the electrocatalysis of $CO_2$, the prepared electrode was pretreated in $CO_2$-saturated 0.5 M $KHCO_3$ by conducting cyclic voltammetry from 0 to −1.0 V vs. RHE with the scan rates of 50 mV s$^{-1}$.

### Physical characterization

X-ray diffraction (XRD) patterns were recorded on X-ray diffractometer (D8 Advanced, Bruker, Germany) equipped with Cu K$_\alpha$ radiation with a scanning rate of 5° min$^{-1}$. The morphologies were observed by scanning electron microscopy (SEM, Apreo S LoVac, FEI, America) and transmission electron microscope (TEM) with an acceleration voltage of 200 kV (Tecnai G2 F20, FEI, America). The local structure and oxygen vacancy were discerned through Aberration-corrected scanning transmission electron microscopy (JEM-ARM200F, JEOL, Japan). The electron spin resonance (EPR) spectra were obtained on JES-FA 200 spectrometer (JEOL, Japan). Surface elements were analyzed by X-ray photoelectron spectroscopy (XPS, Escalab 250, Thermo SCIENTIFIC, America). All energies were referenced to C 1$s$ peaks (284.8 eV) of the surface adventitious carbon. X-ray absorption spectroscopy (XAS) measurements for the Bi L$_3$-edge were performed in fluorescence mode on beamline 20-BM-B with electron energy of 7 GeV and an average current of 100 mA. The radiation was monochromatized by a Si (111) double-crystal monochromator. X-ray absorption near edge structure (XANES) and extended X-ray absorption fine structure (EXAFS) data reduction and analysis were processed by Athena software.

### Electrochemical measurements

The electrochemical performance in $CO_2RR$ was evaluated in H-type cell and flow cell, respectively while electrochemical workstation (CS150H, Corrtest, China) was used as the electricity power. In H-type electrochemical cell, the proton exchange membrane (Nafion 117) was selected to separate cathode and anode while $CO_2$-saturated 0.5 M $KHCO_3$ (50 ML) was filled into the cell as electrolyte. The reference electrode and counter electrode were Ag/AgCl (saturated KCl). and Pt plate ($1 \times 1\ cm^2$), respectively. In flow-cell system, the GDE was prepared by electrodepositing BOC onto commercial carbon paper with hydrophobic microporous layer (Sigracet, Fuel cell store). Specifically, the bare GDL of this carbon paper was covered by Kapton tape to avoid the blockage of microporous channel. Then, the BOC-GDE was

prepared in the same process as described in synthesis section. The flow cell experiments were carried out in a commercial cell (GaossUnion, Tianjin, China). The synthesized BOC-GDE was used as cathode while Ni foam was selected as anode. The reference electrode was Ag/AgCl (saturated KCl) as well. $CO_2$ was fed through the GDL of cathode with the flow rate of 50 mL min$^{-1}$. Both the catholyte and anolyte electrolyte (1 M KOH or 1 M $KHCO_3$) were circulated through a peristaltic pump with the flow rate of 10 mL min$^{-1}$. Linear sweep voltammetry (LSV) measurement was conducted preliminarily to evaluate the activity of catalysts with the scan rates of 5 mV s$^{-1}$. Then, chronoamperometry measurement was adopted to evaluate the selectivity and activity in $CO_2RR$ at one certain potential. In this work, the potentials were converted to RHE scale based on the Nernst equation as follows:

$$E(\text{vs.RHE}) = E(\text{vs.Ag/AgCl}) + 0.197\text{V} + 0.0591 \times \text{pH} \tag{1}$$

The gas product was analyzed by the on-line gas chromatograph (GC-2014, Shimadzu, Japan) equipped with thermal conductivity detector (TCD) and flame ionization detector (FID). The Faradaic efficiency for gas products (including CO and $H_2$) were calculated by the following equation:

$$FE(\%) = \frac{Q_{product}}{Q_{total}} \times 100\% = \frac{zF\upsilon c}{j_{total}V_m} \times 100\% \tag{2}$$

where $Q_{product}$ is the number of electric charge for CO and $H_2$, and $Q_{total}$ represents the total number of electric charge during the whole $CO_2RR$ process. z represents the number of transfer electrons for producing a molecular product, which is 2 and 2 for CO and $H_2$, respectively. $\upsilon$ is the flow rate of gas products through GC and c stands for the concentration of gaseous products. F is the Faraday constant (96485 C mol$^{-1}$) and $V_m$ is the gas molar volume (24 L mol$^{-1}$ at the condition of T = 20°C, $P$ = 101.3 kPa). $j_{total}$ is the recorded total current.

As for liquid product (formate), $^1$H nuclear magnetic resonance (NMR) spectra (Avance III 400 MHz, Bruker, Germany) was used to quantify the production of formate. The Faradaic efficiency of formate was calculated by the following equation:

$$FE(\%) = \frac{Q_{product}}{Q_{total}} \times 100\% = \frac{z_{formate}FN_{formate}}{j_{total}t} \times 100\% \tag{3}$$

where $Q_{product}$ is the number of electric charge for formate and $Q_{total}$ is the total number of electric charge in $CO_2RR$. $z_{formate}$ is the number of transfer electrons for producing a molecular formate, which is 2. $N_{formate}$ represents the moles of formate based on the stand curve line of $^1$H NMR. $j_{total}$ is the total current on average during a fixed time (t) recorded by electrochemical working station and F is the Faraday constant (96485 C mol$^{-1}$).

The turnover frequency (TOF) was calculated according to the equation as follows

$$TOF = \frac{j \times FE_{formate}}{z_{formate}Fn} \tag{4}$$

where j is the total current density; $FE_{formate}$ is the corresponding Faradaic efficiency; z is the number of transfer electrons, which is 2; F is the Faraday constant (96485 C mol$^{-1}$); n is the mole of active sites and all Bi atoms in electrode were assumed to be active sites in our work.

To investigate the effect of water dissociation, $D_2O$ was used to relpace the solvent of $H_2O$ in 0.5 M $KHCO_3$ while the test of $CO_2RR$ was the same with the details above. The liquid products were analyzed by $^1$H NMR and $^2$H NMR to detect the signal of H and D, respectively. Due to the difference in tunneling probability between H and D, the water dissociation barrier can be different, which affects the reaction rate if

water dissociation is involved in the rate-limiting step. To be specific, the reaction of water dissociation will proceed along the potential energy surface in general (Supplementary Fig. 28a, route 1). With the consideration of tunneling probability, the particle will move as marked in red arrow (route 2) due to the probability of matter wave (assuming the wavelength $\lambda$). The tunnelling probability is positive correlation with $\lambda$ while $\lambda_H$ with small nuclear mass is larger than $\lambda_D$ according to de Broglie formula. As a result, the dissociation of $H_2O$ requires smaller energy than that of $D_2O$. If water is involved in RLS, the KIE (the reaction rate ratio between $H_2O$ and $D_2O$) will be >1 and the higher KIE value indicates that water dissociation plays a more important role in rate-limiting step. Apart from the H vs. D tunneling probability, the water dissociation is also influenced by water adsorption and OH adsorption. Herein, the water dissociation can be accelerated by improving OH* adsorption and reducing the energetic barrier of water dissociation due to the introduction of $V_O$. Thus, the difference between the reaction rate in $H_2O$ and $D_2O$ is smaller exhibited as the decreased KIE value.

To investigate the effect of the difference of viscosity between $H_2O$ and $D_2O$, we have executed a finite element based simulation about the diffusions of $CO_2$ (aq) in $H_2O$ and $D_2O$ on planar electrode surfaces. For $H_2O$, the diffusion coefficient is set to be $1.9 \times 10^{-9}$ m$^2$ s$^{-1}$, and the velocity field in convection is set to be $-5 \times 10^{-4}$ m s$^{-1}$. As for the case of $D_2O$, we import the Wilke Chang equation to discuss the deviation in diffusion coefficient with the value of $1.65 \times 10^{-9}$ m$^2$ s$^{-1}$ in terms of the viscosity coefficient[70]. The results of the concentration distribution are shown in Supplementary Fig. 28b, which indicates that the diffusion thickness of BOC in $H_2O$ is slightly larger than that in $D_2O$ (5.00 μm of $H_2O$ vs. 4.74 μm of $D_2O$). As a result, the current difference is estimated to be 5–10% when switching the electrolyte from $H_2O$ to $D_2O$. Besides, the bulk concentration of $CO_2$ in $D_2O$ is a little higher than that in $H_2O$ due to the higher solubility, which will conversely shrink this diffusion current gap between $H_2O$ and $D_2O$. While, the smallest KIE value in our work is of 1.19 which is larger than the diffusion current difference induced by mass transfer. In addition, the KIE experiments were carried out with stirred electrolyte in order to mitigate the limit of mass transport. Overall, the reaction rate ratio ($k_{H2O}/k_{D2O}$) is predominately affected by water dissociation probability due to tunneling effect and the KIE value can be used to reflect the water dissociation probability.

### In situ Raman spectroscopy
The Raman spectra were recorded on Raman spectrometer (XploRa, HORIBA Scientific, Japan) equipped with a green laser of 532 nm. In situ Raman spectra were obtained through the commercial reactor and $CO_2$ was continuously bubbled into the solution of 0.5 M $KHCO_3$ which was pumped into the reactor slowly. The signal was collected at open circuit voltage (OCV) first and different potentials were applied to monitor the stability of BOC. Meanwhile, the laser hit one certain point of electrode in the whole process. It's worth mentioning that the signal of carbonate and $CO_3$* was observed accompanied by Bi = O at 155 cm$^{-1}$ whereas it was difficult to detect these signals at carbon paper without samples and pure electrolyte, which excluded the disturbing effects of minor carbonate in electrolyte. The vibrational frequency of materials with rich $^{13}$C tends to shift towards lower wavenumber according to equation as follows:

$$\frac{\omega_0 - \omega}{\omega_0} = 1 - \left(\frac{12 + c_0}{12 + c}\right)^{\frac{1}{2}} \tag{5}$$

$\omega$ is the frequency of one certain Raman mode in the $^{13}$C-riched materials; $\omega_0$ is the corresponding frequency in the $^{12}$C materials; c is the concentration of $^{13}$C with the value of 0.99 while $c_0$ is the natural abundance of $^{13}$C with the value of 0.0107. For $^{13}CO_3$, $\omega$ can be deduced with the value of 1026 cm$^{-1}$.

### DFT calculations
The DFT calculations are implemented via the Quantum Espresso. Spin-polarized DFT calculations were performed with periodic super-cells under the generalized gradient approximation (GGA) using the Perdew-Burke-Ernzehof (PBE) functional for exchange-correlation and the ultrasoft pseudopotentials for nuclei and core electrons[57]. The kinetic energy and charge-density cutoff by pseudo producers are 30 Ry and 300 Ry, respectively. the convergence criteria are set as $10^{-4}$ Ry/Bohr of Cartesian force components acting on each atom and $10^{-4}$ Ry of total energy. The 001 and 100 planes are cleaved to model the terrace and edge of $Bi_2O_2CO_3$, respectively. The terrace models are consisted with two atoms layers, while edge model contains thickness of 7 Å. During the structure optimization, half of atoms are kept fixed. The chemical potentials of adsorbates X* is calculated by the following expression:

$$\mu_{X^*} = E_{X^*} - E_* + ZPE_{X^*} - TS_{X^*} + \int C^{X^*}_p dT \tag{6}$$

The $E_{X^*}$ and $E_*$ are the DFT based total energies of active sites with and without the adsorbates X*. The ZPE and TS are the contributions from zero points and entropies of adsorbates, whose values are listed on Supplementary Table 8.

### PV-EC system
Nickel and iron co-doped molybdenum oxide ($Mo_{0.9}Ni_{0.05}Fe_{0.05}O_2$) nanospheres reported before was chosen as the anode for oxygen evolution reaction while BOC-2, the best sample in this work, served as cathode to catalyze $CO_2$RR to formate. A commercial three junction GaInP/GaInAs/Ge solar cell (0.846 cm$^2$) was driven by an AM 1.5 solar simulator (XES-50S1-RY, San-Ei Electric, Japan) to provide electricity. The solar to formate conversion efficiency ($\eta$) was calculated as below,

$$\eta(\%) = \frac{P_{out}}{P_{in}} \times 100\% = \frac{J_{solar} \times FE_{formate} \times S_{electrode} \times E^0_{formate}}{P_{solar} \times S_{illuminatedarea}} \times 100\% \tag{7}$$

$J_{solar}$ means the current density in the working system, $S_{electrode}$ is the geometric area of working electrode, $E^0_{formate}$ is thermodynamic cell potential for formate with the value of 1.25 V, $P_{solar}$ is the power of input solar (100 mW cm$^{-2}$) and $S_{illuminated\,area}$ is the irradiation area of PV.

## Data availability
All relevant data are available from the corresponding author on reasonable request.

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

## Acknowledgements

This work was financially supported by the National Natural Science Foundation of China (22072106, L.L.), National Key R&D Program of China (2021YFA1500800, L.L.), Science Foundation of State Key Laboratory of Structural Chemistry (20210025, J.C.), Science Foundation of Fujian Province (2021J01526, J.C.) and Beiyang Reserved Academic Program of Tianjin University (L.L.).

## Author contributions

X.C. carried out the experiments and related data processing of this work. J.C. conducted theoretical calculations. H.C. and Q.Z. gave help in PV-EC system. J.L. performed EPR measurements. J.W.C. and Y.S. performed XRD tests. X.C. and L.L wrote and revised the manuscript. D.W. and J.Y. provided experimental supports. L.L. supervised the project.

## Competing interests

The authors declare no competing interests.
