## [Peer review file · Nature Communications]

REVIEWER COMMENTS

Reviewer #1 (Remarks to the Author):

In this study, Liu et.al report the introduction of oxygen vacancies on a $\text{Bi}_2\text{O}_2\text{CO}_3$ catalyst to promote formate production from CO_2RR . The authors hypothesize that oxygen vacancies accelerate water dissociation kinetics and increase the kinetics of CO_2RR by providing protons.

Engineering oxygen vacancies on metal catalysts to promote CO_2 electrolysis to formate is not a new technique and previous studies have shown this approach for Sn and Bi based catalysts. However, a detailed mechanistic study has been performed in this study to unravel that the rate limiting step is the water dissociation reaction which is interesting and important for CO_2RR . Hence, I recommend the publication of this article provided the authors address some major concerns addressed below:

1. The authors have reported different techniques such as kinetic isotope effect (KIE) over the different catalysts to show that water dissociation kinetics is altered with VO engineered samples. However, cations have been shown to play a major role in water dissociation kinetics (pubs.acs.org/doi/10.1021/jacs.1c10171) with weakly hydrated cations showing higher activity. To further prove their hypothesis, the authors should report the performance of HER for BOC-C and BOC-2 samples in a different electrolyte such as NaHCO_3 and discuss the interactions between cations and water molecule.

2. In Line 158 and 159: The authors state that, “ There is no difference in diameter of semicircle over BOC with different Vo contents”.

a) However, LSV curves in Fig.2 clearly show that BOC-2 is having superior CO_2RR performance as the authors state in Line 116. Wouldn't the charge transfer resistance of BOC-2 be lower than the other catalysts? Fig S16 reports only the Nyquist plots of all the samples but further data and discussion is necessary.

b) The authors should provide details on frequency range and amplitude used for EIS analysis and fit the data with an equivalent circuit model. The uncompensated resistance, charge transfer resistance and capacitance values can be calculated from this and discussion should be made based on these numbers for all 5 samples.

3. ECSA results of all samples must be provided to rule out any effects of surface area on enhanced performance.

4. Fig. 6 (Supplementary) and Table 1 (Supplementary) : The authors show and quantify oxygen vacancy by XPS analysis. However, there have been reports on the wrong use of attributing O1s signal at 531-532 eV to oxygen vacancies (<https://doi.org/10.1016/j.susc.2021.121894>). The article states that ex-situ XPS measurements of oxygen are not possible. The authors should clarify this and provide additional characterization technique (EPS or XAFS) to show the oxygen vacancies.

5. In Fig.1k, there are no significant differences in the intensity of Bi 4f for BOC-C and the four BOC with VO engineered samples. What is the reason for this? Shouldn't the VO engineered samples have a relatively higher percentage of Bi⁺/Bi than Bi³⁺ compared to that of BOC-C? The authors should explain this.

6. From EXAFS study reported, the authors should calculate and report the coordination number of Bi³⁺ and oxygen atoms for the 4 samples and compare it with bare BOC sample to further verify the differences in oxygen vacancies.

7. In Line 118, the authors state: "the selectivity and activity as function of potentials were testified by chronopotentiometry". Product selectivity from CO₂RR are calculated through in-line gas chromatography and NMR spectroscopy for liquid products. Here, details on how the faradaic efficiency (FE) of formate, CO and H₂ were quantified are missing. Sample calculation of formate from NMR and CO and H₂ from GC analysis must be reported.

8. Fig.S17: The authors show proton NMR spectra for formate. If possible, the authors must include ¹³C NMR results as well.

9. BET surface area measurements for all the samples must be provided to identify the differences in surface area and porosity of the samples.

10. Line 143-144: Details on the experimental setup for flow-cell system using gas diffusion electrode (GDE) are not reported. The authors should provide details on how the catalysts were prepared and deposited on a carbon GDL. In addition, details on CO₂ flowrate and stability of GDE must be reported at 200 mA/cm².

11. Additionally, double layer capacitance measurements must be provided for flow cell tests to report ECSA and turn over frequency of catalysts. This data must be compared with previous literature reports on other GDE catalysts (Sn, In) for formate production at high current densities (although the authors

have compared the values in literature for H-cells in Table S3 and S4). Discussion on why BOC-2 might be beneficial at these high current densities should be added.

12. For the GDE tests in flow cells, wouldn't the BOC-VO samples reduce to metallic Bi at these high current densities? If that's the case, how does the performance of BOC-2 compare with bare BOC-C in these flow cells at high current densities? The authors should report the performance of bare BOC-C sample as well in flow cells at high current densities to discuss if there are any significant differences or if the observed differences in formate production for BOC-C and BOC-2 occurs only in H-cells. This is because at current densities $> 50\text{mA/cm}^2$, the local $\text{pH} > 12$ which might alter the water dissociation kinetics. Discussion on this aspect must be included.

13. Since the authors study the correlation between oxygen vacancy (VO) and formate production, providing additional numbers on optimal Vo concentration for a given electrode size and current density would be a useful metric.

14. Previous studies on BiO_2CO_3 as catalyst for CO_2RR to formate should be cited. One example: pubs.acs.org/doi/10.1021/acsomega.7b00437

Minor comments

1. In Fig.2(a), the colors used for the plotting LSV curves in Ar saturated and CO_2 saturated electrolyte solution can be improved and using thicker lines will make it easier for the reader to clearly see the differences for the 5 samples.

2. Line 74-75: Missing "in" in the sentence: "Theoretical analysis shows CO_3 participates the formation of formate as the key surface species"

Reviewer #2 (Remarks to the Author):

All comments are enclosed in the pdf file attached.

Journal	Nature Communications
Number	
Title	Promoting water dissociation for efficient solar driven CO ₂ electroreduction via improving hydroxyl adsorption
Authors	Xin Chen & Lequan Liu et al
Abstract	Exploring efficient electrocatalysts with fundamental understanding of the reaction mechanism is imperative in CO ₂ electroreduction. However, the impact of sluggish water dissociation as proton source and the surface species in reaction are still unclear. Herein, we report a strategy of promoting protonation in CO ₂ RR by implementing oxygen vacancy (VO) engineering on Bi ₂ O ₂ CO ₃ over which high Faradaic efficiency of formate (above 90%) and large partial current density (152 mA cm ⁻²) are achieved. Systematic study reveals that the production rate of formate is mainly hampered by water dissociation, while the introduction of VO accelerates water dissociation kinetics by strengthening hydroxyl adsorption and reduces the energetic span of CO ₂ RR. Moreover, carbonate involved in formate formation as the key surface species is clearly identified by ESR measurements and designed in situ Raman spectroscopy study combined with isotopic labelling. Coupled with photovoltaic device, the solar to formate energy conversion efficiency reaches as high as 13.3%.
Recommendation	Reconsider after major revisions

Referee report to the authors

General comments:

The authors report the metal oxide type material Bi₂O₂CO₃ as CO₂RR catalyst. By tuning the oxygen vacancy degree, the authors were able to steer the catalytic activity and FE of the material for CO₂ to formate conversion.

Material synthesis and characterisation seem to be carried out with care. The authors convincingly showed that they achieved generating Bi₂O₂CO₃ with varying degrees of V_O. Specifically, the BOC-2 sample shows competitive CO₂RR performance (as stand-alone electrocatalyst and integrated in a PEC system) compared to state-of-the-art but not outstanding. Moreover, Bi-type metal oxide catalysts for CO₂RR are known. This reviewer is no expert on x ray-based characterisation.

This reviewer appreciates that the authors have undertaken a detailed investigation of the CO₂RR mechanism to rationalise the different activity and FE of the BOC materials with different V_O content. The overall presentation of gathered experimental data and its discussion is well done. *However, interpretation of individual data set seems to be overstretched. Clarification and more critical assessment of the data sets is needed to support the mechanistic claims.*

Overall, the direct link of the role of water dissociation to CO₂RR activity is of interest for the scientific community. If the manuscript could be revised to include a critical assessment of

the experimental data sets that supports the mechanistic interpretation, it should be published in Nat Comm.

Specific comments:

Comment 1: EIS analysis

EIS spectroscopy has been carried out to investigate the effect of V₂O₅ on the conductivity and hence resistivity of the material (SI Figure S16). In fact, the EIS data is significantly different between BOC-1 to BOC-4. The interpretation that the conductivity of the materials did not change is not valid. Also, the EIS data indicates that the electrochemical interface of the material is different. This should have an impact on electrochemical reactions proceeding at the interface of the respective material. A more thorough discussion of the data is therefore needed to account for these differences.

First, the plot SI Figure 16 lacks the BOC-C, which is cardinal to allow for comparison with the unmodified material. Furthermore, the x axis should also be expanded to show the behaviour at the higher resistances.

Second, the resistances of the dielectric layer are shifted to different values. Also, the circles do not fully close, particularly for the BOC-1 material, which behaves completely different to the other materials.

From the EIS data, it becomes evident that the materials do show a change in electrochemical interfacial properties beyond the mere introduction of O-deficient binding sites. This should be critically discussed and incorporated into the mechanistic model.

Comment 2: Tafel slope analysis

Tafel plots are shown in Fig. 3a. The authors stated that the BOC materials exhibit different Tafel slopes and concluded that BOC-2 exhibiting the lowest slope correlating to the fastest CO₂RR activity.

This reviewer is not sure that the correlation between a low Tafel slope corresponding to faster kinetics can be drawn. This could be true but only under certain conditions. A proper citation (page 7, line 160-162) is missing

Noteworthy, only 3 points were collected to derive the Tafel slopes. Hence, the values for the slopes are affected by large errors, and it is not possible to use the values for a mechanistic model. Moreover, error bars are not shown for the individual current values.

Importantly, the absolute Tafel slope values should also be discussed. The different absolute values could indicate a change in catalytic mechanism as discussed in the literature. Also, the overall value of 200-400 mV/dec seems to be off compared to typically reported values.

Comment 3: KIE analysis

The authors further investigated the KIE by exchanging H₂O with D₂O. Also, they showed that H₂O and not HCO₃⁻ is the major proton source for CO₂RR.

While all samples show interesting KIEs and pH dependences, the interpretation of the data seems flawed to this reviewer. The authors argued that a KIE of close to 1 indicates that water dissociation is likely not affecting the catalytic kinetics (which means that H₂O desorption is not rate-limiting), backed up with 1 citation. This reviewer doesn't think that this rationale is correct. The exchange H₂O to D₂O affects a range of properties, which is well known in the literature. Besides the mentioned dissociation effect, H vs D tunnelling probability is the most significant change. The high KIE effect therefore likely reflects proton tunnelling to be rate-limiting, which is also corroborated by the pH-dependence experiments of the authors.

Additionally, D₂O has an intrinsically different pD value than H₂O, which should be considered in the discussion. Also, D₂O shows a 20% higher viscosity which likely affects the overall kinetics by affecting mass transport, constitution of the Helmholtz layer etc.

All these properties should be included in a discussion before the conclusion of H₂O dissociation to be the driver for the KIE could be made.

Comment 4: In-situ Raman studies

The authors conducted in situ Raman studies to investigate the fate of the material during catalysis. The overall Raman experiment is well designed. Unfortunately, the Raman spectra are a bit noisy and overall low in intensity, making it hard to unambiguously assign bands and changes. The discussion of the data is vague and important statements are missing for the reader to understand and follow the rationale. In total, this reviewer believes that the presented in situ Raman spectra and particularly the discussion of the carbonate bands do not conclusively show CO₃ adsorption.

4.1 It would be good to also show the higher frequency region from 1200-1700 cm⁻¹ to monitor the changes ongoing with in the C-O region if possible. In this region, other intermediates, such as Bi-OCO etc., may be observed. The high frequency region holds therefore a lot of information that aids to support the mechanistic model To rule out such other pathways, the high frequency region may support the mechanistic model.

4.2 The authors concluded the stability of the BOC materials from the largely unaltered Raman band structure upon going from OCV to cathodic potentials. This seems to be correct. However, the signal intensity is decreased at cathodic potentials. Therefore, the authors should present the Raman spectra at OCV after the potential series to demonstrate that the original spectrum can be re-obtained.

4.3 The interpretation of the ¹²CO₂ and ¹³CO₂ experimental Raman spectra is not fully clear to this reviewer. On page 11 line 250, the authors state that adding ¹²CO₂ results in an ¹²CO₃ peak. In Figure 5c, the spectral region is marked in blue. The black spectrum shows the intrinsic BOC exhibiting the band at 1067 cm⁻¹ originating from CO₃ groups of the BOC. But when ¹²CO₂ is added only one band is visible (OCV pale blue spectrum) which is low frequency shifted. If the lower frequency band is the ¹²CO₃ species, where is the band for the intrinsic CO₃ of the BOC? Also, it should be clarified how much time was waited between the spectra in ¹²CO₂ and ¹³CO₂. The authors should also clarify that the exchange does not occur at potentials lower than the onset potentials.

4.4 Same question as in **4.2** arises when ¹³CO₂ is introduced. The intrinsic CO₃ band of BOC is missing. The rising lower frequency band at 1012 cm⁻¹ may originate from ¹³CO₃ formed. How can the authors be sure that this reflects carbonate formed at the material interface/ at the material. Purging CO₂ into KHCO₃ 0.5 M should change the bicarbonate/carbonate equilibrium and hence affects the carbonate band. Fig 5b shows the Raman spectrum of the electrolyte lacking any carbonate/bicarbonate bands, which is unusual given the high concentration of 0.5 M. To demonstrate that the 1067 cm⁻¹ bands are coming from solely from the material interaction, the authors should show the Raman spectra in Ar-saturated electrolyte under potential in 0.5 M KHCO₃ and in KPHO₄ (same potentials as shown in Fig 5b). The reason is that it is not clear where the carbonate bands are coming from, i.e., electrolyte saturation or material or both, and why the material's intrinsic carbonate is not observed.

4.5 This reviewer appreciates the fact that the comparison to BOC-3 and BOC-C has been made. However, the interpretation of Fig 5e is not clear. BOC-C seems to exhibit partial

exchange. The authors should discuss this and what this means in the context of their hypothesis.

Reviewer #3 (Remarks to the Author):

The authors investigate the effect of oxygen vacancies in Bi₂O₂CO₃ (BOC) on enhancing the CO₂ reduction reaction to formate by boosting the water dissociation. The experimental results and theoretical calculation demonstrate that oxygen vacancies stabilize the hydroxyl group and drive efficient water dissociation. Consequently, the energy barrier for *OCHO intermediate is reduced on oxygen sites of BOC. This work is done well and clearly shows the evidence for their claim. However, the reviewer can find one major inconsistency in BOC electrocatalysts compared to the previous publications and minor issues that should be thoroughly addressed via major revision.

- 1) In Figure S1, there are several XRD peaks. BOC nanosheets appear to be polycrystalline rather than single crystalline.
- 2) The reviewer cannot find the defect in Figures 1d and S5. Please explain how to identify one.
- 3) In Figure 1g, the y-axis values show less than 3% difference in intensity that might be within the experimental error. Also, why the presence of oxygen vacancy shows a larger intensity in the y-axis than that of oxygen sites?
- 4) Why does controlling the water:ethylene glycol ratio change the oxygen vacancies concentration in BOC? A brief description should be at least provided.
- 5) If the water dissociation is promoted, hydrogen evolution reaction (HER) might also be enhanced due to the increase in the number of protons near the surface. So, please discuss why the competing HER is not enhanced but only CO₂ reduction is enhanced.
- 6) Authors claim that “there is no obvious change in phase structure, morphology and valance state, which demonstrates the stability of BOC-2 (Supplementary Fig. 15).” However, this is in great contradiction with the recently reported BOC catalyst (<https://doi.org/10.1002/eem2.12490>). Based on the Pourbaix diagram of BOC, it should be reduced to Bi under cathodic bias ([https://doi.org/10.1016/S0378-7753\(00\)00640-6](https://doi.org/10.1016/S0378-7753(00)00640-6)). Authors should clearly state the reason for the discrepancy by citing the above references.
- 7) The reviewer also wants to know and recommend explaining why H₂O showed a higher formate production rate compared to D₂O in Figures 3b and 3c.

Response to the reviewers' comments

Contents

1. Response to Reviewer 1's comments.....	2
2. Response to Reviewer 2's comments.....	38
3. Response to Reviewer 3's comments.....	63

Reviewer #1 (Remarks to the Author):

In this study, Liu et.al report the introduction of oxygen vacancies on a $\text{Bi}_2\text{O}_2\text{CO}_3$ catalyst to promote formate production from CO_2RR . The authors hypothesize that oxygen vacancies accelerate water dissociation kinetics and increase the kinetics of CO_2RR by providing protons.

Engineering oxygen vacancies on metal catalysts to promote CO_2 electrolysis to formate is not a new technique and previous studies have shown this approach for Sn and Bi based catalysts. However, a detailed mechanistic study has been performed in this study to unravel that the rate limiting step is the water dissociation reaction which is interesting and important for CO_2RR . Hence, I recommend the publication of this article provided the authors address some major concerns addressed below:

Response: We are grateful to the reviewer for endorsing the research points of our work and raising serious, insightful comments. These comments are valuable for improving the quality of our work. We have studied these suggestions carefully and supplementary experiments were carried out. Please see below our point-by point responses.

1. The authors have reported different techniques such as kinetic isotope effect (KIE) over the different catalysts to show that water dissociation kinetics is altered with V_{O} engineered samples. However, cations have been shown to play a major role in water dissociation kinetics (pubs.acs.org/doi/10.1021/jacs.1c10171) with weakly hydrated cations showing higher activity. To further prove their hypothesis, the authors should report the performance of HER for BOC-C and BOC-2 samples in a different electrolyte such as NaHCO_3 and discuss the interactions between cations and water molecule.

Response: We thank the reviewer for this comment. The HER performance of BOC-C and BOC-2 were evaluated specifically in Ar-saturated NaHCO_3 solutions as suggested. As can be seen from Figure R1, the overpotential of BOC is lower in KHCO_3 than that in NaHCO_3 , which is consistent with the result reported by Koper et al. and indicates that cation is indeed effective in water dissociation (*J. Am. Chem. Soc.* **144**, 1589–1602 (2022); this work has been cited in our revised manuscript as Ref. 61). However, less enhancement induced by cations is observed on BOC-2, which can be attributed to the intrinsic faster water dissociation kinetics due to V_{O} introduction.

Meanwhile, in addition to KIE, the remarkable enhancement of water dissociation in CO_2RR promoted by V_{O} is also validated at different local pH while the cation concentration is the same. As shown in Figure R2, the formate production rate of BOC-2 with optimized V_{O} concentration is much higher. These results demonstrate that the introduction of V_{O} is able to accelerate water dissociation kinetics

which is in favor of formate production as a result.

Changes have been made in the revised manuscript (Page 9, Line 192):

"... These results agree well with the formate production rates discussed above and a strong correlation among V_O , water dissociation ability and formate production is identified. Besides, less enhancement in activity induced by cations can be observed on BOC-2 as compared with that on BOC-C, indicating the intrinsic faster water dissociation kinetics of BOC-2 (Supplementary Fig.30)⁶¹. The role of V_O in water dissociation was further clarified through varying local pH achieved by using three electrolytes with different buffer capacity (K_2HPO_4 , K_2CO_3 and K_2SO_4). It has been demonstrated that the local pH value increased in the order of K_2HPO_4 , K_2CO_3 and K_2SO_4 , while water dissociation becomes more difficult at high pH value¹⁸. The same concentrations of cations among those three kinds of electrolyte can exclude the effect of cations in CO_2RR ⁶²⁻⁶³. ..."

Figure R1. The LSV curves of BOC-C and BOC-2 in Ar-saturated 0.5 M $NaHCO_3$ and 0.5 M $KHCO_3$, respectively (corresponding to Supplementary Fig.30 in the revised Supplementary Information).

Figure R2. a KIE experiments with 0.5 M $\text{KHCO}_3\text{-H}_2\text{O}$ and 0.5 M $\text{KHCO}_3\text{-D}_2\text{O}$ as electrolyte. b The production rates of formate over BOC-1, BOC-2, BOC-3 and BOC-4 in 0.25 M K_2HPO_4 , K_2CO_3 and K_2SO_4 electrolytes, where the concentration of cations are the same in all the electrolytes.

2. In Line 158 and 159: The authors state that, “There is no difference in diameter of semicircle over BOC with different V_{O} contents”.

a) However, LSV curves in Fig.2 clearly show that BOC-2 is having superior CO_2RR performance as the authors state in Line 116. Wouldn't the charge transfer resistance of BOC-2 be lower than the other catalysts? Fig S16 reports only the Nyquist plots of all the samples but further data and discussion is necessary.

b) The authors should provide details on frequency range and amplitude used for EIS analysis and fit the data with an equivalent circuit model. The uncompensated resistance, charge transfer resistance and capacitance values can be calculated from this and discussion should be made based on these numbers for all 5 samples.

Response: We thank the reviewer for this important comment and apologize for the imprecise expression in original manuscript. To understand the electrochemical interface and the role of V_{O} thoroughly, the EIS measurement of BOC-C was implemented complementarily with the frequency range of 1-1000000 Hz and amplitude of 5 mV. All the EIS data were fitted with equivalent circuit consisting of two sections based on previous studies (*Nano Energy* **71**, 104653 (2020); *Angew. Chem. Int. Ed.* **61**, e202206077 (2022)), over which the fitted results match well with experimental data. As shown in Figure R3, the first section represents the uncompensated solution resistance (R_s) while the second section reflects the interfacial properties including charge transfer resistance (R_{ct}) and the resistance of adsorbed intermediates including water and hydroxyl (R_p). The equivalent circuit parameters are calculated and summarized in Table R1.

With the introduction of V_{O} , the charge transfer of BOC-2 is accelerated with smaller R_{ct} value as compared with that of BOC-C, which can be in favor of reducing the overpotential and increasing the conductivity in CO_2RR . This deduction is supported by the results of performance evaluation during CO_2RR (Figure R4). However, the introduction of excessive V_{O} (BOC-4) limits the charge transfer rate, which may be resulted from the structure distortion.

In addition, it can be noted that the resistance induced by water and hydroxyl adsorption (R_p) is related with V_{O} concentration, which implies that V_{O} is able to change water and hydroxyl adsorption. According to our KIE experiments and DFT analysis (Figure R5), the introduction of V_{O} can strengthen hydroxyl adsorption and expediate water dissociation kinetics, which contributes to reducing the energetic span in formate formation. Overall, the role of V_{O} in BOC can be concluded to promote water

dissociation accompanied by accelerating charge transfer, which is beneficial for reducing the overpotential and boosting formation production rate in CO₂RR.

Changes have been made in the revised manuscript (Page 7, Line 170):

"... Next, electrochemical impedance spectroscopy (EIS) was carried out and fitted by equivalent circuit to investigate the electrochemical interface properties⁵⁸⁻⁵⁹. With the introduction of V_O, the charge transfer can be accelerated, which is beneficial for conductivity and reducing the overpotential (Supplementary Fig. 25 and Supplementary Table 6). Besides, the resistance induced by water and hydroxyl adsorption (R_p) is related with V_O concentration, implying that V_O might affect the proton transfer by changing water and hydroxyl adsorption. Before conducting the investigation on proton transfer further, the proton source was clarified through isotopic labelling of D₂O. ..."

Figure R3. The EIS plot of different samples and the inset is the employed equivalent circuit. The circles represent the EIS data while the solid line represent corresponding fitted results (corresponding to Supplementary Fig.25 in the revised Supplementary Information).

Figure R4. The LSV curves of different samples in CO₂-saturated (solid line) and Ar-saturated (dashed line) 0.5 M KHCO₃ (corresponding to Fig. 2a in the revised manuscript).

Figure R5. The role of V_O in water dissociation. a The KIE value of H/D for different samples. The red and black line represent samples before and after annealing, respectively. b The free energy of step III in formate formation, Δμ_{OH-w}, the adsorption energy of OH* and H₂O*.

Table R1. The equivalent circuit parameters in EIS analysis (corresponding to Supplementary Table 6 in the revised Supplementary Information).

Sample	R _s (Ω)	R _{ct} (Ω)	R _p (Ω)
C-BOC	6.8	48.4	398.7
BOC-1	6.3	42.9	366.9
BOC-2	7.8	40.0	348.5
BOC-3	7.2	46.7	380.0
BOC-4	6.5	46.5	399.8

3. ECSA results of all samples must be provided to rule out any effects of surface area on enhanced performance.

Response: We thank the reviewer for this valuable comment. To obtain the value of electrochemical surface area (ECSA), CV measurements at different scan rates were carried out at first to derive the

value of double layer capacitance (C_{dl}) as shown in Figure R6. Then, the ECSA of different samples were calculated based on the following equation:

$$ECSA = S \times R_f = S \times \frac{C_{dl}}{C_s}$$

Among this, S stands for the real surface area of smooth electrode, equivalent to the geometric area of work electrode (1 cm² in our case). R_f is the roughness factor, which is equal to the ratio of C_{dl} and general specific capacitance of metal oxide (C_s , 60 $\mu\text{F cm}^{-2}$, *Angew. Chem. Int. Ed.* **59**, 10807–1081 (2020); *ACS Catal.* **10**, 743–750 (2020)). The corresponding values of C_{dl} and ECSA for different samples are summarized in Table R2. It can be found that the introduction of V_O slightly increases the ECSA, which should be helpful for activity.

To clarify the effect of ECSA and get more insight into the intrinsic activity of BOC with V_O , the partial current of formate of different samples were normalized by ECSA. As can be seen from Figure R7, the $j_{\text{formate}} \text{ ECSA normalized}$ of BOC-2 is much larger than that of BOC-C, indicating the introduction of V_O is notably in favor of elevating the intrinsic activity of BOC apart from the increased ECSA.

Changes have been made in the revised manuscript (Page 7, Line 161):

"... To clarify the intrinsic activity of BOC with V_O , the j_{formate} of different samples were normalized by specific surface area (Supplementary Fig. 21 and 22) and electrochemical surface area, respectively (Supplementary Fig. 23 and Supplementary Table 5). It can be found that the introduction of V_O notably increases the intrinsic activity and the degree of this enhancement is closely related with the V_O contents (Supplementary Fig. 24). ..."

Figure R6. CV curves and the derived C_{dl} value of different catalysts (corresponding to Supplementary Fig. 23 in the revised Supplementary Information).

Figure R7. The partial current density of formate normalized by ECSA for different catalysts (corresponding to Supplementary Fig. 24a in the revised Supplementary Information).

Table R2. The C_{dl} value and electrochemical surface area (ECSA) for BOC-C, BOC-1, BOC-2, BOC-3 and BOC-4 (corresponding to Supplementary Table 5 in the revised Supplementary Information).

	BOC-C	BOC-1	BOC-2	BOC-3	BOC-4
C_{dl} ($\mu\text{F cm}^{-2}$)	12.8	17.5	19.3	16.4	17.0
ECSA (cm^2)	0.21	0.29	0.32	0.27	0.28

4. Fig. 6 (Supplementary) and Table 1 (Supplementary): The authors show and quantify oxygen vacancy by XPS analysis. However, there have been reports on the wrong use of attributing O1s signal at 531-532 eV to oxygen vacancies (<https://doi.org/10.1016/j.susc.2021.121894>). The article states that ex-situ XPS measurements of oxygen are not possible. The authors should clarify this and provide additional characterization technique (EPS or XAFS) to show the oxygen vacancies.

Response: We thank the reviewer for this valuable comment and read the related work carefully. We agree with the reviewer that the direct measurement of V_O from XPS measurements is impossible. The signal at 531-532 eV is ascribed to the hydroxyl adsorption due to V_O . As pointed out by Tapilin et al., hydroxyl will be adsorbed spontaneously on V_O in the presence of water (*J. Phys. Chem. C* **114**, 3609–3613 (2010)). Therefore, it can be deduced that there might be V_O if the signal of hydroxyl appears in XPS. That is, the signal of hydroxyl adsorption in XPS can act as an indicator of V_O existence to some extent.

In order to clarify the presence of V_O further, electron paramagnetic resonance spectroscopy (EPR) measurement was carried out as it is considered as an effective method to detect V_O based on its sensitivity on unpaired electrons. As shown in Figure R8, the signal with the $g=2.002$ which is the characteristic g value of V_O can be observed, demonstrating the presence of V_O strongly. Compared with BOC-C, the intensity of this typical signal increases in the sequence of BOC-1, BOC-2, BOC-3 and BOC-4, which suggests that the BOC with a series of V_O concentration were obtained. The V_O concentration was subsequently quantified through external standard method while 2,2,6,6-tetramethylpiperidinyloxy (TEMPO) was used as standard (*Appl. Catal. B Environ.* **224**, 612–620 (2018); *Sol. RRL* **4**, 2000037 (2020)) and the atomic V_O concentration were estimated to be 0.031%, 0.044%, 0.060%, 0.076% for BOC-1, BOC-2, BOC-3 and BOC-4, respectively (Figure R9 and Table R3).

With the purpose of getting in-depth information about the V_O location and coordinated structure, X-ray absorption spectroscopy including EXAFS and XANES was carried out. The coordination number of Bi-O for BOC with V_O manifests a decrease with the rise of V_O content, which suggests the presence of unsaturated Bi atoms induced by V_O and coincides with our STEM results (Figure R10 and Table R4). Besides, according to the Bi- L_3 XANES edge (Figure R11), the absorption edge of BOC-2 slightly shifts to lower energy compared with that of BOC-1, indicating the lower valance state of Bi induced by V_O formation. Overall, the presence of V_O can be confirmed by XPS, TEM, EPR and XAS characterizations.

Changes have been made in the revised manuscript (Page 5, Line 98):

"... The regular variation of the sharp signal intensity with g value of 2.002 in EPR characterization not only further clarifies the presence of V_O but also indicates that V_O concentration

increases in the sequence of BOC-1, BOC-2, BOC-3 and BOC-4 (Fig. 1h)⁴⁹. Commercial BOC (denoted as BOC-C) was adopted for comparison, and the V_O concentration is much lower than that of BOC-1 as suggested by the weak EPR signal. The V_O concentration was also quantified from EPR while atomic V_O contents are estimated to be 0.031%, 0.044%, 0.060% and 0.076% for BOC-1, BOC-2, BOC-3 and BOC-4, respectively (Supplementary Table 1)⁵⁰. The location of V_O is disclosed from extended X-ray absorption fine structure (EXAFS) spectroscopy, and two peaks at around 1.6 Å and 3.5 Å corresponding to the scattering path of Bi-O and Bi-O-C are found (Fig. 1i, Supplementary Fig. 7 and Supplementary Table 2) ...".

Figure R8. EPR spectra of different samples.

Figure R9. EPR spectrum of TEMPO which is the standard in order to quantify the V_O contents.

Figure R10. Fourier transform of Bi L₃ edge EXAFS data of BOC-1, BOC-2 and BOC-3 recorded at R space (corresponding to Supplementary Fig. 7 in the revised Supplementary Information).

Figure R11. Normalized XANES spectra of Bi foil, BOC-1, BOC-2 and BOC-3. Inset: the magnified area marked by red line (corresponding to Fig. 1j in the revised manuscript).

Table R3. The V_O contents of different samples based on EPR measurements (corresponding to Supplementary Table 1 in the revised Supplementary Information).

Sample	BOC-1	BOC-2	BOC-3	BOC-4
V _O (at.%)	0.031	0.044	0.060	0.076

Table R4. EXAFS fitting parameters at the Bi L₃ edge for BOC-1, BOC-2 and BOC-3. CN: coordination numbers; R: bond distance; σ²: Debye-Waller factors; ΔE₀: the inner potential correction. R factor:

goodness of fit (corresponding to Supplementary Table 2 in the revised Supplementary Information).

Sample	Shell	CN	R(Å)	σ^2	ΔE_0	R factor
BOC-1	Bi-O	2.7	2.23	0.0024	-2.4	0.022
BOC-2	Bi-O	2.6	2.25	0.0026	-1.7	0.023
BOC-3	Bi-O	2.4	2.26	0.0037	-4.4	0.024

6. In Fig. 1k, there are no significant differences in the intensity of Bi 4f for BOC-C and the four BOC with V_O engineered samples. What is the reason for this? Shouldn't the V_O engineered samples have a relatively higher percentage of Bi^{+}/Bi than Bi^{3+} compared to that of BOC-C? The authors should explain this.

Response: We thank the reviewer for this comment. As we discussed in our original manuscript, the peaks of Bi 4f for BOC with V_O shift towards lower binding energy compared with those for BOC-C, indicating the lower valenced state of Bi induced by V_O (*Nano Lett.* **18**, 7372–7377 (2018); *Nano-Micro Lett.* **14** 90 (2022)). Generally, the contents of the certain element can be semi-quantified by relative intensity in XPS spectra rather than the absolute intensity which is influenced by many factors such as surface finish of the sample, the status of XPS equipment, the chemical environment of elements and so on.

With the kind mind of this reviewer, the peaks in Bi 4f are re-fitted, and the peaks of 164.6 and 159.3 are assigned to Bi^{3+} while the peaks of 164.0 and 158.5 correspond to $Bi^{(3-x)+}$ (*Ceram. Int.* **48**, 22163–22171 (2022)). As can be seen from Figure R12, the intensity of $Bi^{(3-x)+}$ rises with the increase of V_O concentration, indicating a relatively higher ratio of lower valence state of Bi due to the introduction of V_O . Besides, the XANES absorption edge of BOC with V_O shifts towards lower binding energy, which implies that BOC with V_O has relative lower valence state on average and matches with the XPS results discussed above (Figure R13).

Figure R12. XPS spectra of Bi 4f for BOC-C, BOC-1, BOC-2, BOC-3 and BOC-4, respectively. The red area represents the peak of $\text{Bi}^{(3-x)+}$ while the green area is the peak of Bi^{3+} (corresponding to Fig. 1k in the revised manuscript).

Figure R13. Normalized XANES spectra of Bi foil, BOC-1, BOC-2 and BOC-3. Inset: the magnified area marked by red line (corresponding to Fig. 1j in the revised manuscript).

6. From EXAFS study reported, the authors should calculate and report the coordination number of

Bi³⁺ and oxygen atoms for the 4 samples and compare it with bare BOC sample to further verify the differences in oxygen vacancies.

Response: We thank the reviewer for this comment. We are really sorry that the XAS data of BOC-4 and BOC-C are in absence in revision due to the limitation of operation period of XAS which is affected by the equipment maintenance schedule and Covid-19 prevention. Fortunately, the BOC-1 was complementarily characterized and the corresponding coordination numbers of Bi-O for BOC-1, BOC-2 and BOC-3 were calculated and summarized in Figure R14a and Table R5. As V_O concentration increases, the coordination number of Bi-O decreases accompanied with the increase of Bi-O bonding length, which can be attributed to the absence of V_O . However, the variation of coordinated number and bonding length are subtle, which indicates that the modification of coordinated environment of Bi induced by V_O is relatively limited.

On the other hand, the clarification of V_O through EXAFS is generally based on the variation of chemical state and coordination number of Bi. Relatively, EPR, as a sensitive technique in detecting the unpaired electron left by V_O , is more direct in qualifying and quantifying V_O . Compared with BOC-C, the signal of V_O increases in the order of BOC-1, BOC-2, BOC-3 and BOC-4 (Figure R14b). Apart from that, XPS is reliable in analyzing the surface chemical state of Bi while the BOC with V_O show lower chemical state than BOC-C, indicating the presence of V_O in our prepared BOC samples (Figure R14c). Even though the XAS data of BOC-C and BOC-4 are in absence, we believe that the BOC samples with different V_O concentrations are successfully prepared according to STEM, EPR, XPS and available XAS results. Thanks a lot for your understanding.

Changes have been made in the revised manuscript (Page 5, Line 104):

“... The location of V_O is disclosed from extended X-ray absorption fine structure (EXAFS) spectroscopy, and two peaks at around 1.6 Å and 3.5 Å corresponding to the scattering path of Bi-O and Bi-O-C are found (Fig. 1i, Supplementary Fig. 7 and Supplementary Table 2). The intensity of Bi-O manifests a decrease for BOC with more V_O contents while there is no obvious difference in that of Bi-O-C, which implies that V_O mainly exists in Bi-O-Bi structure⁵¹. X-ray absorption near edge structure (XANES) region (Fig. 1j) shows that the absorption edge for BOC with V_O just slightly shifts to lower energy, which is in agreement with XPS spectra of Bi 4f and indicates the presence of low chemical state Bi induced by V_O (Fig. 1k)⁵². Based on the characterizations above, it can be concluded that the BOC with different contents of V_O is successfully prepared. ...”

Figure R14. A Fourier transform of Bi L₃ edge EXAFS data at R space (corresponding to Supplementary Fig. 7 in the revised Supplementary Information), b EPR spectra and c XPS spectra of all the BOC samples (corresponding to Fig. 1k in the revised manuscript).

Table R5. EXAFS fitting parameters at the Bi L₃ edge for BOC-1, BOC-2 and BOC-3. CN: coordination numbers; R: bond distance; σ^2 : Debye-Waller factors; ΔE_0 : the inner potential correction. R factor: goodness of fit (corresponding to Supplementary Table 2 in the revised Supplementary Information).

Sample	Shell	CN	R(Å)	σ^2	ΔE_0	R factor
BOC-1	Bi-O	2.7	2.23	0.0024	-2.4	0.022
BOC-2	Bi-O	2.6	2.25	0.0026	-1.7	0.023
BOC-3	Bi-O	2.4	2.26	0.0037	-4.4	0.024

7. In Line 118, the authors state: “the selectivity and activity as function of potentials were testified by chronopotentiometry”. Product selectivity from CO₂RR is calculated through in-line gas chromatography and NMR spectroscopy for liquid products. Here, details on how the faradaic efficiency (FE) of formate, CO and H₂ were quantified are missing. Sample calculation of formate from NMR and CO and H₂ from GC analysis must be reported.

Response: Thank you for this kind reminder. The gas products from CO₂RR were analyzed by a gas chromatograph (GC-2014, Shimadzu, Japan) equipped with thermal conductivity detector (TCD) and flame ionization detector (FID). The Faradaic efficiency for gas products (including CO and H₂) were

calculated by the following equation:

$$FE (\%) = \frac{Q_{\text{product}}}{Q_{\text{total}}} \times 100\% = \frac{zFvc}{j_{\text{total}}V_m} \times 100\%$$

where Q_{product} is the number of electric charge for CO and H₂, and Q_{total} represents the total number of electric charge during the whole CO₂RR process. z represents the number of transfer electrons for producing a molecular product, which is 2 and 2 for CO and H₂, respectively. v is the flow rate of gas products through GC and c stands for the concentration of gaseous products. F is the Faraday constant (96485 C mol⁻¹) and V_m is the gas molar volume (24 L mol⁻¹ at the condition of T=20°C, P=101.3 kPa). j_{total} is the recorded total current.

As for liquid product (formate), ¹H nuclear magnetic resonance (NMR) spectra (Avance III 400 MHz, Bruker, Germany) was used to quantify the production of formate. The Faradaic efficiency of formate was calculated by the following equation:

$$FE (\%) = \frac{Q_{\text{product}}}{Q_{\text{total}}} \times 100\% = \frac{z_{\text{formate}}FN_{\text{formate}}}{j_{\text{total}}t} \times 100\%$$

where Q_{product} is the number of electric charge for formate and Q_{total} is the total number of electric charge in CO₂RR. z_{formate} is the number of transfer electrons for producing a molecular formate, which is 2. N_{formate} represents the moles of formate based on the stand curve line of ¹H NMR. j_{total} is the total current on average during a fixed time (t) recorded by electrochemical working station and F is the Faraday constant (96485 C mol⁻¹).

Changes have been made in the revised manuscript (Page 16, Line 369):

"... With regard to reflecting the CO₂RR activity of different catalysts, chronoamperometry was performed at a series of potentials for at least an hour. The gas-phase products were analyzed online every 20 minutes by a gas chromatograph (GC-2014, Shimadzu, Japan) equipped with thermal conductivity detector (TCD) and flame ionization detector (FID). The Faradaic efficiency for gas products (including CO and H₂) were calculated by the following equation:

$$FE (\%) = \frac{Q_{\text{product}}}{Q_{\text{total}}} \times 100\% = \frac{zFvc}{j_{\text{total}}V_m} \times 100\%$$

where Q_{product} is the number of electric charge for CO and H₂, and Q_{total} represents the total number of electric charge during the whole CO₂RR process. z represents the number of transfer electrons for producing a molecular product, which is 2 and 2 for CO and H₂, respectively. v is the flow rate of gas products through GC and c stands for the concentration of gaseous products. F is the Faraday constant (96485 C mol⁻¹) and V_m is the gas molar volume (24 L mol⁻¹ at the condition of T=20°C, P=101.3 kPa). j_{total} is the recorded total current.

As for liquid product (formate), ¹H nuclear magnetic resonance (NMR) spectra (Avance III 400 MHz, Bruker, Germany) was used to quantify the production of formate. In detail, 400 μL of electrolyte after reaction was collected and mixed with 100 μL of D₂O and DMSO after dilution (100 ppm) as the

internal stand. The concentration of formate was calculated based on the calibration curve from a series of concentrations of standard HCOOK solutions. The Faradaic efficiency of formate was calculated by the following equation:

$$\text{FE (\%)} = \frac{Q_{\text{product}}}{Q_{\text{total}}} \times 100\% = \frac{z_{\text{formate}} F N_{\text{formate}}}{j_{\text{total}} t} \times 100\%$$

where Q_{product} is the number of electric charge for formate and Q_{total} is the total number of electric charge in CO₂RR. z_{formate} is the number of transfer electrons for producing a molecular formate, which is 2. N_{formate} represents the moles of formate based on the stand curve ~~line~~ of ¹H NMR. j_{total} is the total current on average during a fixed time (t) recorded by electrochemical working station and F is the Faraday constant (96485 C mol⁻¹). ..."

8. Fig.S17: The authors show proton NMR spectra for formate. If possible, the authors must include ¹³C NMR results as well.

Response: We are thankful to this reviewer's comment and the ¹³C NMR spectra were obtained with ¹³CO₂ as the feed gas during CO₂RR (Figure R15). The signal at 172 ppm is attributed to H¹³COO⁻, demonstrating that the formate is indeed originate from CO₂.

Changes have been made in revised manuscript (Page 7, Line 154):

"... Moreover, ¹³CO₂ labeling experiment was carried out. The proton doublet resulting from H-¹³C coupling and H¹³COO⁻ is observed in ¹H NMR and ¹³C NMR, respectively (Supplementary Fig. 19), demonstrating that the produced formate derives from CO₂⁵³. ..."

Figure R15. The ^{13}C NMR spectra with $^{13}\text{CO}_2$ as carbon source during CO_2RR (corresponding to Supplementary Fig. 19c in the revised Supplementary Information).

9. BET surface area measurements for all the samples must be provided to identify the differences in surface area and porosity of the samples.

Response: Thanks for this valuable comment. Considering the porous structure of BOC, N_2 adsorption/desorption isotherms were supplemented to study the specific surface area and pore size distribution of different samples. As shown in Figure R16, the BOC with different V_{O} contents show similar specific surface area while the values of the BOC samples synthesized in our work are a little larger as compared with BOC-C. Besides, all the BOC samples possess micropores and mesoporous pores with similar pore size distribution (Figure R17).

To clarify the effect of BET surface in the enhancement of formate production, the j_{formate} curves were normalized by the BET surface area, where the intrinsic activity of BOC-2 is notably higher compared with that of other samples (Figure R18). This result demonstrates the specific surface area has negligible effect on the difference of formate production rate in our work.

Changes have been made in the revised manuscript (Page 7, Line 161):

"... To clarify the intrinsic activity of BOC with V_{O} , the j_{formate} of different samples were normalized by specific surface area (Supplementary Fig. 21 and 22) and electrochemical surface area, respectively (Supplementary Fig. 23 and Supplementary Table 5). It can be found that the introduction of V_{O} notably increases the intrinsic activity of and the degree of this enhancement is closely related with the V_{O} contents (Supplementary Fig. 24). ..."

Figure R16. N₂ adsorption/desorption isotherms of a BOC-C, b BOC-1, c BOC-2, d BOC-3 and e BOC-4. The corresponding BET surface are listed in the plot (corresponding to Supplementary Fig. 21 in the revised Supplementary Information).

Figure R17. Pore size distributions of a BOC-C, b BOC-1, c BOC-2, d BOC-3 and e BOC-4 (corresponding to Supplementary Fig. 22 in the revised Supplementary Information).

Figure R18. The j_{formate} normalized by the specific surface area of all the catalysts at different negative potentials (corresponding to Supplementary Fig. 24b in the revised Supplementary Information).

10. Line 143-144: Details on the experimental setup for flow-cell system using gas diffusion electrode (GDE) are not reported. The authors should provide details on how the catalysts were prepared and deposited on a carbon GDL. In addition, details on CO₂ flowrate and stability of GDE must be reported at 200 mA/cm².

Response: We appreciate the reviewer's helpful comment. The GDE was prepared by electrodepositing BOC onto commercial carbon paper with hydrophobic microporous layer (Sigracet, Fuel cell store). Specifically, the bare GDL of this carbon paper was covered by Kapton tape to avoid the blockage of microporous channel. Then, the BOC-GDE was prepared in the same process as described in methods section. The flow cell experiments were carried out in a commercial cell (GaossUnion, Tianjin, China). The synthesized BOC-GDE was used as cathode while Ni foam was selected as anode. CO₂ was fed through the GDL of cathode with the flow rate of 50 mL min⁻¹. Both the catholyte and anolyte electrolyte (1 M KOH or 1 M KHCO₃) were circulated through a peristaltic pump with a flow rate of 10 mL min⁻¹. These details on experimental setup and catalyst preparation have been added in the revised manuscript.

To evaluate the stability of BOC-2 at high current density, the performance of BOC-2 was further tested at the current density of 200 mA cm⁻² in 1 M KOH and 1 M KHCO₃, respectively (Figure R19). The FE_{formate} of BOC-2 is above 80% during 15 h with 1 M KOH as electrolyte. To relieve the carbonate formation and KOH consumption, KHCO₃ was used as electrolyte at the same set-up, and FE_{formate} is above 60% during 15 h. The decreased selectivity of formate in KHCO₃ can be explained by the relatively low pH which is favorable for hydrogen generation. Overall, the BOC-2 exhibits good stability at industrial current density, demonstrating the potential in real applications.

Changes have been made in the revised manuscript (Page 7, Line 150):

"... In addition, a stable FE_{formate} of 80% at large current density of 200 mA cm⁻² can be observed during 15 h, which indicates the potential for practical application (Supplementary Fig. 18). ..."

(Page 17, Line 393):

"... In flow-cell system, the GDE was prepared by electrodepositing BOC onto commercial carbon paper with hydrophobic microporous layer (Sigracet, Fuel cell store). Specifically, the bare GDL of this carbon paper was covered by Kapton tape to avoid the blockage of microporous channel. Then, the BOC-GDE was prepared in the same process as described in synthesis section. The flow cell experiments were carried out in a commercial cell (GaossUnion, Tianjin, China). The synthesized BOC-GDE was used as cathode while Ni foam was selected as anode. CO₂ was fed through the GDL of cathode with the flow rate of 50 mL min⁻¹. Both the catholyte and anolyte electrolyte (1 M KOH or 1 M KHCO₃) were circulated through a peristaltic pump with the flow rate of 10 mL min⁻¹. ..."

Figure R19. The stability test at the current density of 200 mA cm^{-2} with a 1 M KOH and b 1 M KHCO₃ as electrolyte, respectively (corresponding to Supplementary Fig.18 in the revised Supplementary Information).

11. Additionally, double layer capacitance measurements must be provided for flow cell tests to report ECSA and turn over frequency of catalysts. This data must be compared with previous literature reports on other GDE catalysts (Sn, In) for formate production at high current densities (although the authors have compared the values in literature for H-cells in Table S3 and S4). Discussion on why BOC-2 might be beneficial at these high current densities should be added.

Response: We thank the reviewer for this comment. The C_{dl} was measured in flow cell with 1 M KOH as electrolyte (Figure R20). As summarized in Table R6, the ECSA of BOC-2 is larger than that of BOC-C, confirming the introduction of V_O contributes to the increased ECSA. Besides, it can be noted that the ECSA of all the samples in flow cell is larger than that in H-cell, which may be originated from the sufficient contact between electrode and electrolyte in alkaline electrolyte as pointed out by Zhuang et al. (*Nat. Energy* 7, 835–843 (2022))

The turnover frequency (TOF) of BOC-2 was calculated to be 0.72 s^{-1} at 200 mA cm^{-2} according to the equation as follows:

$$\text{TOF} = \frac{j \times \text{FE}_{\text{formate}}}{zFn}$$

where j is the total current density; $\text{FE}_{\text{formate}}$ is the corresponding Faradaic efficiency; z is the number of transfer electrons, which is 2; F is the Faraday constant (96485 C mol^{-1}); n is the mole of active sites and all Bi atoms were assumed to be active sites here.

In light of the reviewer's suggestion, the performance of our BOC-2 in flow-cell system was further compared with other reported electrocatalysts in formate production including Bi-based, Sn-based and In-based materials as summarized in Table R7. It's worth noting that there are a few works with reported TOF value in flow cell as far as we known. Therefore, other important works with outstanding performance in flow-cell are also listed even though the TOF values are in absence. As can be seen from Table R7, the BOC-2 exhibits good performance at high current density, which can be attributed to the superior water dissociation on BOC-2 with V_{O} introduction. Due to the higher current density at more negative potentials, the protons are consumed rapidly and the local pH of electrode can be alkaline which will make water dissociation more difficult. However, the introduction of V_{O} has been demonstrated in our work to be notably in favor of water dissociation by reducing the energetic barrier through strengthening hydroxyl adsorption, which promotes formate production. Detailed investigation about the effect of V_{O} at high current density is discussed in comment 12.

Changes have been made in the revised manuscript (Page 7, Line 146):

"... To relieve the limitation on mass transfer and pursue commercial current density requirements, BOC-2 was further integrated into gas diffusion electrode (GDE) and evaluated in flow-cell system⁵³. The j_{formate} of BOC-2 is 1.6 times as high as that of BOC-C at -1.68 V while the turnover frequency can be up to 0.72 s^{-1} at 200 mA cm^{-2} , which demonstrates the activity enhancement through introducing V_{O} can also be achieved in flow-cell system (Supplementary Fig. 17 and Supplementary Table 4). ..."

(Page 9, Line 200):

"... As can be seen from Fig. 3d and Supplementary Fig. 31, the production rate of formate for BOC-2 is much higher than that of other samples at high local pH value, which indicates the beneficial effect of V_{O} . Moreover, it can be found that BOC-2 shows superior activity and selectivity than BOC-C in flow-cell at large current density where water dissociation is more difficult due to the high local pH induced by rapid protons depletion (Supplementary Fig.17 c, d). So, it can be concluded that water dissociation is involved in the rate-determining step for CO_2RR to formate, and the presence of V_{O} remarkably boosts the performance of CO_2RR by accelerating H_2O dissociation. ..."

Figure R20. CV curves and the derived C_{dl} value of different catalysts as GDEs.

Table R6. The electrochemical surface area of all catalysts as GDEs.

	BOC-C	BOC-1	BOC-2	BOC-3	BOC-4
C_{dl} ($\mu\text{F cm}^{-2}$)	57.5	70.9	71.9	62.7	67.5
ECSA (cm^2)	0.96	1.18	1.20	1.05	1.13

Table R7. Electrochemical performance for CO₂RR to formate in flow-cell reported recently (corresponding to Supplementary Table 4 in the revised Supplementary Information).

Catalyst	Electrolyte	Current density (mA cm ⁻²)	TOF (s ⁻¹)	FE _{formate} (%)	Ref.
Sn-N-C	0.5 M KHCO ₃	200	1.4	50-60	[25]
BiO _n cluster	0.1 M KOH (MEA)	500	4.2	90	[26]
Bi ₂ O ₃ /BiO ₂	0.5 M KHCO ₃	114	0.27	98.12	[27]
BOC with V _O	1.0 M KOH	200	0.72	81	This work
SnO ₂ NPs	1.0 M KOH	147	/	46	[28]
BOC with V _O	1.0 M KOH	1000	/	93	[18]
Bi _{0.1} Sn	1.0 M KOH	200	/	97	[29]
Bi ₂ O ₃ @C	1.0 M KOH	224	/	93	[30]
H-Sn ₃ O ₄	1.0 M KOH	462	/	91	[31]
SnO ₂ /Sn	1.0 M KOH	114	/	94	[32]
Sn-In NPs	1.0 M KOH	236	/	94	[33]
SnS	1.0 M KOH	120	/	88	[34]
Bi-In alloy NPs	1.0 M KOH	250	/	97.8	[35]
MIL-68(In)- NH ₂	1.0 M KHCO ₃	114	/	94.4	[36]

ZnIn ₂ S ₄	1.0 M	300	/	99.3	[37]
	KHCO ₃				

The turnover frequency (TOF) of BOC-2 was calculated according to the equation as follows

$$\text{TOF} = \frac{j \times \text{FE}_{\text{formate}}}{zFn}$$

where j is the total current density; $\text{FE}_{\text{formate}}$ is the corresponding Faradaic efficiency; z is the number of transfer electrons, which is 2; F is the Faraday constant (96485 C mol^{-1}); n is the mole of active sites and all Bi atoms in electrode were assumed to be active sites in our work. The n is calculated based on the equation as follows:

$$n = \frac{x_{\text{Bi}} \times m_{\text{cat.}}}{M_{\text{Bi}}}$$

where, x_{Bi} is the metallic composition in electrocatalyst, $m_{\text{cat.}}$ is the mass of electrocatalyst and M_{Bi} is the relative atomic mass of Bi. The TOF cited in this Table is calculated based on this equation.

For Ref 25 and 26, the metallic composition is 8.8 wt% and 8.0 wt%, respectively, according to ICP-MS results. In our work, the metallic composition is considered as 82.0 wt% based on chemical formula.

Reference

- "...
18. Fan, T. et al. Achieving high current density for electrocatalytic reduction of CO₂ to formate on bismuth-based catalysts. *Cell Rep. Phys. Sci.* **2**,100353 (2021).
- ...
25. Duarte, M. et al. Enhanced CO₂ electroreduction with metal-nitrogen-doped carbons in a continuous flow reactor. *J. CO₂ Util.* **50**, 101583 (2021).

26. Jiang, X. et al. Boosting CO₂ electroreduction to formate via bismuth oxide clusters. *Nano Res.* 1-8 (2022) DOI: 10.1007/s12274-022-5073-0.
27. Feng, X. et al. Bi₂O₃/BiO₂ nanoheterojunction for highly efficient electrocatalytic CO₂ reduction to formate. *Nano Lett.* **22**, 1656-1664 (2022).
28. Liang, C. et al. High efficiency electrochemical reduction of CO₂ beyond the two-electron transfer pathway on grain boundary rich ultra-small SnO₂ nanoparticles. *J. Mater. Chem. A.* **6**, 10313 (2018).
29. Li, L. et al. Stable, active CO₂ reduction to formate via redox modulated stabilization of active sites. *Nat. Commun.* **12**, 5223 (2021).
30. Deng, P. et al. Metal-organic framework-derived carbon nanorods encapsulating bismuth oxides for rapid and selective CO₂ electroreduction to formate. *Angew. Chem. Int. Ed.* **59**, 10807-10813 (2020).
31. Liu, L. et al. Tuning Sn₃O₄ for CO₂ reduction to formate with ultra-high current density. *Nano Energy* **77**, 105296 (2020).
32. Ning, S. et al. Electrochemical reduction of SnO₂ to Sn from the bottom: in-situ formation of SnO₂/Sn heterostructure for highly efficient electrochemical reduction of carbon dioxide to formate. *J. Catal.* **399**, 67-74 (2021).
33. Wang, J. et al. In-Sn alloy core-shell nanoparticles: In-doped SnO_x shell enables high stability and activity towards selective formate production from electrochemical reduction of CO₂. *Appl. Catal. B: Environ.* **288**, 119979 (2021).
34. Zou, J., Lee, C., Wallace, G. Boosting formate production from CO₂ at high current densities over a wide electrochemical potential window on a SnS catalyst. *Adv. Sci.* **8**, 2004521 (2021).

35. Yao, K. et al. Metal-organic framework derived dual-metal sites for electroreduction of carbon dioxide to HCOOH. *Appl. Catal. B: Environ.* **311**, 121377 (2022).

36. Wang, Z. et al. Efficient electroconversion of carbon dioxide to formate by a reconstructed amino-functionalized indium-organic framework electrocatalyst. *Angew. Chem. Int. Ed.* **60**, 19107–19112 (2021).

..."

12. For the GDE tests in flow cells, wouldn't the BOC- V_0 samples reduce to metallic Bi at these high current densities? If that's the case, how does the performance of BOC-2 compare with bare BOC-C in these flow cells at high current densities? The authors should report the performance of bare BOC-C sample as well in flow cells at high current densities to discuss if there are any significant differences or if the observed differences in formate production for BOC-C and BOC-2 occurs only in H-cells. This is because at current densities $> 50\text{mA/cm}^2$, the local pH > 12 which might alter the water dissociation kinetics. Discussion on this aspect must be included.

Response: We thank the reviewer for this important comment about the stability. To monitor the stability of bulk phase of BOC-2, XRD measurement was conducted on BOC-2 after the stability test at 200 mA cm^{-2} (Figure R21), where the structure of BOC-2 can be well retained without the presence of metallic Bi (Figure R22). Then, the in-situ Raman measurements were carried out for BOC-2 at a series of negative potentials to investigate the change on surface of BOC and there is no characteristic peak of metallic Bi during the whole process. These results provide strong evidence on the stability of BOC-2 (Figure R23).

Even though Bi-based oxide may be reduced at negative potentials in thermodynamics, the stability on BOC-2 can be attributed to the exothermic CO_2 adsorption (Figure R24) and the high local pH where the oxide state of Bi is stable (*Cell Rep. Phys. Sci.* **2**, 100353 (2021); *Nat. Commun.* **12**, 5223 (2021)). That is, the Bi^{3+} tends to be reduced to metallic Bi in thermodynamics at negative potentials while Bi^0 is easily to be oxidized to BOC during CO_2RR due to high local pH and accessible CO_2 adsorption.

In light of the reviewer's suggestions, the performance of BOC-C and BOC-2 were studied additionally in flow cell to clarify the role of V_0 at high current density. As can be seen from Figure R25, the BOC-2 exhibits larger partial current density of formate compared with BOC-C at more negative potentials and the j_{formate} of BOC-2 is 1.6 times as high as that of BOC-C at -1.68 V . As far as we known, the kinetics of water dissociation will be sluggish due to quick consumption of protons at

high current density accompanied by high local pH, which will limit the reaction rate of CO₂RR. Even though, BOC-2 with V_O still exhibits superior ability in formate production, which can be attributed to the improved water dissociation induced by V_O. Similar results can be observed by applying a series of current density on electrode, where BOC-2 shows higher FE_{formate} relative to BOC-C (Figure R25d). Thus, the enhancement in activity through introducing V_O is also notable at high current density, beneficial from the accelerated water dissociation kinetics.

Besides, the effect of local pH was investigated in our original manuscript by using three kinds of solutions with different buffer capacity (K₂HPO₄, K₂CO₃ and K₂SO₄) as electrolytes, which is responsible for varying different local pH around electrode. The production rate of formate for BOC-2 is much higher at high local pH environment (Figure R26). Overall, BOC-2 with V_O exhibits superior activity in formate production compared with BOC-C at high current density where water dissociation is more difficult, which confirms that the introduction of V_O can promote formate formation by accelerating water dissociation kinetics.

Changes have been made in the revised manuscript (Page 7, Line 150):

“... In addition, a stable FE_{formate} of 80% at large current density of 200 mA cm⁻² can be observed during 15 h, which indicates the potential for practical application (Supplementary Fig. 18). Even though Bi³⁺ could be reduced at negative potentials in thermodynamics, the good stability of BOC can be explained by the spontaneous CO₂ adsorption and high local pH where the oxide state of Bi is stable^{29, 33, 54-55}. ...”

(Page 9, Line 200):

“... As can be seen from Fig. 3d and Supplementary Fig. 31, the production rate of formate for BOC-2 is much higher than that of other samples at high local pH value, which indicates the beneficial effect of V_O. Moreover, it can be found that BOC-2 shows superior activity and selectivity than BOC-C in flow-cell at large current density where water dissociation is more difficult due to the high local pH induced by rapid protons depletion (Supplementary Fig. 17 c, d). So, it can be concluded that water dissociation is involved in the rate-determining step for CO₂RR to formate, and the presence of V_O remarkably boosts the performance of CO₂RR by accelerating H₂O dissociation. ...”

Figure R21. The stability test at the current density of 200 mA cm^{-2} (corresponding to Supplementary Fig.18 in the revised Supplementary Information).

Figure R22. XRD patterns of BOC-2 as GDE in flow cell after stability test (corresponding to Supplementary Fig.18c in the revised Supplementary Information).

Figure R23. The in-situ Raman spectra of BOC-2 with a series of negative potentials applied (corresponding to Fig. 5b in the revised manuscript).

Figure R24. The free energy of CO₂ adsorption which is exothermic.

Figure R25. The FE_{formate} of a BOC-2 and b BOC-C in flow-cell systems at different negative potentials. c the partial current density of BOC-C and BOC-2 in flow-cell. d The FE_{formate} of BOC-C and BOC-2 at different current density (corresponding to Supplementary Fig. 17 in the revised Supplementary Information).

Figure R26. The production rates of formate over BOC-1, BOC-2, BOC-3 and BOC-4 in K_2HPO_4 , K_2CO_3 and K_2SO_4 electrolytes.

13. Since the authors study the correlation between oxygen vacancy (V_O) and formate production, providing additional numbers on optimal V_O concentration for a given electrode size and current density would be a useful metric.

Response: Thanks for this valuable comment. In revision, the V_O concentration was further obtained through EPR measurements with 2,2,6,6-tetramethylpiperidinyloxy (TEMPO) as standard (*Appl. Catal. B Environ.* **224**, 612–620 (2018); *Sol. RRL* **4**, 2000037 (2020), Figure R27). The molar ratio of V_O in BOC-2 which is the optimal sample in our work with the area of 1 cm^2 and high formate production, is estimated to be 0.044% (Table R8).

Changes have been made in the revised manuscript (Page 5, Line 98):

"... The regular variation of the sharp signal intensity with g value of 2.002 in EPR characterization not only further clarifies the presence of V_O but also indicates that V_O concentration increases in the sequence of BOC-1, BOC-2, BOC-3 and BOC-4 (Fig. 1h)⁴⁹. Commercial BOC (denoted as BOC-C) was adopted for comparison, and the V_O concentration is much lower than that of BOC-1 as suggested by the weak EPR signal. The V_O concentration was also quantified from EPR while atomic V_O contents are estimated to be 0.031%, 0.044%, 0.060% and 0.076% for BOC-1, BOC-2, BOC-3 and BOC-4, respectively (Supplementary Table 1)⁵⁰. The location of V_O is disclosed from extended X-ray absorption fine structure (EXAFS) spectroscopy, and two peaks at around 1.6 Å and 3.5 Å corresponding to the scattering path of Bi-O and Bi-O-C are found (Fig. 1i, Supplementary Fig. 7 and Supplementary Table 2) ...".

Figure R27. EPR spectrum of TEMPO which was used as the standard to quantify the concentration of

V_O in BOC samples.

Table R8. The V_O contents of different samples based on EPR measurements (corresponding to Supplementary Table 1 in the revised Supplementary Information).

Sample	BOC-1	BOC-2	BOC-3	BOC-4
V _O (at.%)	0.031	0.044	0.060	0.076

14. Previous studies on Bi₂O₂CO₃ as catalyst for CO₂RR to formate should be cited. One example: pubs.acs.org/doi/10.1021/acsomega.7b00437

Response: We thank the reviewer's kind remind. We really appreciate these excellent works about Bi₂O₂CO₃ (*ACS Omega* **2**, 2561-2567 (2017); *Angew. Chem. Int. Ed.* **57**, 13283-13287 (2018); *Chem. Commun.* **55**, 12392 (2019); *Sustain. Energy Fuels*, **4**, 2831-2840 (2020); *J. Mater. Chem. A* **8**, 24486 (2020)) which gave us lots of inspiration to implement our work and were included in our supplementary information in our previous version. To emphasize the importance of these studies, they are cited in our revised text.

Changes have been made in the revised manuscript (Page 2, Line 50):

"... For electrode materials, metal oxide attracts broad attentions due to high selectivity and low overpotentials which are the crucial parameters for commercial scale in CO₂RR to formate²⁰⁻²⁸. Moreover, ..."

(Page 3, Line 62):

"...Previous studies propose that CO₂ tends to adsorb on oxygen site in metal oxide to form CO₃* and participates in sequent reduction process as the key surface species, which is considered to be the origin of attractive performance for metal oxide electrocatalysts³⁹⁻⁴⁵. ..."

References

"...

25. Zhang, Y. et al. Controllable synthesis of few-layer bismuth subcarbonate by electrochemical exfoliation for enhanced CO₂ reduction performance. *Angew. Chem. Int. Ed.* **57**, 13283-13287 (2018).

26. Liu, P., Zu, M., Zheng, L. & Yang, H. Bismuth oxyiodide microflower-derived catalysts for efficient CO₂ electroreduction in a wide negative potential region. *Chem. Commun.* **55**, 12392 (2019).

27. An, X. et al. The in situ morphology transformation of bismuth-based catalysts for the effective electroreduction of carbon dioxide. *Sustain. Energy Fuels.* **4**, 2831-2840 (2020).

28. Yuan, W. et al. In situ transformation of bismuth metal–organic frameworks for efficient selective electroreduction of CO₂ to formate. *J. Mater. Chem. A* **8**, 24486 (2020).

...

39. Lv, W. et al. Bi₂O₂CO₃ nanosheets as electrocatalysts for selective reduction of CO₂ to formate at low overpotential. *ACS Omega* **2**, 2561-2567 (2017).

..."

Minor comments

1. In Fig.2(a), the colors used for the plotting LSV curves in Ar saturated and CO₂ saturated electrolyte solution can be improved and using thicker lines will make it easier for the reader to clearly see the differences for the 5 samples.

Response: We appreciate the reviewer’s kind suggestion and the LSV curves in Figure 2a are now plotted by using thicker lines to distinguish the activity of different samples more clearly.

Figure R28. The LSV curves of different samples in CO₂-saturated (solid line) and Ar-saturated (dashed line) 0.5 M KHCO₃ (corresponding to Fig. 2a in the revised manuscript).

2. Line 74-75: Missing “in” in the sentence: “Theoretical analysis shows CO₃ participates the formation of formate as the key surface species”

Response: We thank the reviewer for this comment and we have corrected this ignorance in our revised manuscript.

Changes have been made in the revised manuscript (Page 3, Line 74):

"... Theoretical analysis shows CO_3^* participates **in** the formation of formate as the key surface species, which is demonstrated clearly through electron spin resonance (ESR) measurements and in situ Raman spectroscopy combined with isotopic labelling. Finally, full-cell electrocatalysis coupled with solar cell was constructed and achieves the solar to formate energy conversion efficiency of 13.3%. ..."

Reviewer 2

The authors report the metal oxide type material $\text{Bi}_2\text{O}_2\text{CO}_3$ as CO_2RR catalyst. By tuning the oxygen vacancy degree, the authors were able to steer the catalytic activity and FE of the material for CO_2 to formate conversion. Material synthesis and characterisation seem to be carried out with care. The authors convincingly showed that they achieved generating $\text{Bi}_2\text{O}_2\text{CO}_3$ with varying degrees of V_{O} . Specifically, the BOC-2 sample shows competitive CO_2RR performance (as stand-alone electrocatalyst and integrated in a PEC system) compared to state-of-the-art but not outstanding. Moreover, Bi-type metal oxide catalysts for CO_2RR are known. This reviewer is no expert on x ray-based characterisation.

This reviewer appreciates that the authors have undertaken a detailed investigation of the CO_2RR mechanism to rationalise the different activity and FE of the BOC materials with different V_{O} content. The overall presentation of gathered experimental data and its discussion is well done. However, interpretation of individual data set seems to be overstretched. Clarification and more critical assessment of the data sets is needed to support the mechanistic claims.

Overall, the direct link of the role of water dissociation to CO_2RR activity is of interest for the scientific community. If the manuscript could be revised to include a critical assessment of the experimental data sets that supports the mechanistic interpretation, it should be published in Nat Comm.

Response: We really appreciate the reviewer's great endeavors to review our work. These positive and constructive comments could help us get thorough insight into our work. In this revision, additional experiments and discussion are included based on the reviewer's suggestions and concerns.

Specific comments:

Comment 1: EIS analysis

EIS spectroscopy has been carried out to investigate the effect of V_{O} on the conductivity and hence resistivity of the material (SI Figure S16). In fact, the EIS data is significantly different between BOC-1 to BOC-4. The interpretation that the conductivity of the materials did not change is not valid. Also, the EIS data indicates that the electrochemical interface of the material is different. This should have an impact on electrochemical reactions proceeding at the interface of the respective material. A more thorough discussion of the data is therefore needed to account for these differences. First, the plot SI Figure 16 lacks the BOC-C, which is cardinal to allow for comparison with the unmodified material. Furthermore, the x axis should also be expanded to show the behaviour at the higher resistances. Second, the resistances of the dielectric layer are shifted to different values. Also, the circles do not fully close, particularly for the BOC-1 material, which behaves completely different to the other materials. From the EIS data, it becomes evident that the materials do show a change in electrochemical interfacial properties beyond the mere introduction of O-deficient binding sites. This should be critically

discussed and incorporated into the mechanistic model.

Response: We thank the reviewer for his or her insightful suggestion and apologize for the imprecise expression in original manuscript. Under the reviewer's guidance, the EIS data of BOC-C was added and the x axis was expanded to analysis the EIS data more thoroughly as shown in Figure R29 and Table R9. The arc shape of BOC-1 can be attributed to the dispersion effect due to the roughness, inhomogeneity and porous structure of stacked nanosheets.

In order to investigate the electrochemical interface properties, the equivalent circuit was employed to fit the impedance result as shown in Figure R29, over which the fitted results match well with experimental data. The model used here includes two sections: (1) the first corresponds to the uncompensated solution resistance (R_s) while there are some variations in R_s for different samples owing to the roughness and experimental error such as electrode area, distance, temperature and so on; (2) the second section reflects interfacial resistance at electrode-electrolyte interface including R_{ct} (charge transfer process) and R_p (water and hydroxyl adsorption). With the introduction of V_O , the charge transfer resistance of BOC-2 is lower than that of BOC-C, which indicates that V_O is able to optimize the conductivity of BOC and accelerate the charge transfer rate. This faster charge transfer is conducive to the lower overpotential of BOC with V_O as shown in Figure R30. However, BOC with much more V_O (BOC-4) shows larger R_{ct} , which might be resulted from the structure distortion.

Apart from the variation in R_{ct} , it can be noted that the R_p of different samples also exhibits the correlation with V_O concentration, which hints that V_O is effective in water and hydroxyl adsorption. To figure out this specific effect on water and hydroxyl adsorption induced by introducing V_O , DFT analysis and KIE experiments were carried out. As shown in Figure R31, the presence of V_O is able to strengthen hydroxyl adsorption and promotes water dissociation which is involved in the rate-limiting process in formate formation. As a result, BOC with V_O exhibits higher formate production rate which is beneficial from the reduced energetic span in CO_2RR . Overall, the introduction of V_O plays a crucial role in modifying the electrochemical interface property through strengthening hydroxyl adsorption accompanied by accelerating charge transfer, which is conducive in promoting water dissociation and boosting CO_2RR as a result.

Changes have been made in the revised manuscript (Page 7, Line 170):

"... Next, electrochemical impedance spectroscopy (EIS) was carried out and fitted by equivalent circuit to investigate the electrochemical interface properties⁵⁹⁻⁶⁰. With the introduction of V_O , the charge transfer can be accelerated, which is beneficial for conductivity and reducing the overpotential (Supplementary Fig. 25 and Supplementary Table 6). Besides, the resistance induced by water and hydroxyl adsorption (R_p) is related with V_O concentration, implying that V_O might affect the proton transfer by changing water and hydroxyl adsorption. ..."

Figure R29. The EIS plot of different samples and the inset is the employed equivalent circuit. The circles represent the EIS data while the solid line represent corresponding fitted results (corresponding to Supplementary Fig.25 in the revised Supplementary Information).

Figure R30. The LSV curves of different samples in CO_2 -saturated (solid line) and Ar-saturated (dashed line) 0.5 M KHCO_3 (corresponding to Fig 2a in the revised manuscript).

Figure R31. The role of V_O in water dissociation. a The KIE value of H/D for different samples. The red and black line represent samples before and after annealing, respectively. b The free energy of step III in formate formation, $\Delta\mu_{OH-w}$, the adsorption energy of OH^* and H_2O^* .

Table R9. The equivalent circuit parameters in EIS analysis (corresponding to Supplementary Table 6 in the revised Supplementary Information).

Sample	R_s (Ω)	R_{ct} (Ω)	R_p (Ω)
C-BOC	6.8	48.4	398.7
BOC-1	6.3	42.9	366.9
BOC-2	7.8	40.0	348.5
BOC-3	7.2	46.7	380.0
BOC-4	6.5	46.5	399.8

Comment 2: Tafel slope analysis

Tafel plots are shown in Fig. 3a. The authors stated that the BOC materials exhibit different Tafel slopes and concluded that BOC-2 exhibiting the lowest slope correlating to the fastest CO_2RR activity. This reviewer is not sure that the correlation between a low Tafel slope corresponding to faster kinetics can be drawn. This could be true but only under certain conditions. A proper citation (page 7, line 160-162) is missing. Noteworthy, only 3 points were collected to derive the Tafel slopes. Hence, the values for the slopes are affected by large errors, and it is not possible use the values for a mechanistic model. Moreover, error bars are not shown for the individual current values.

Importantly, the absolute Tafel slope values should also be discussed. The different absolute values could indicate a change in catalytic mechanism as discussed in the literature. Also, the overall value of

200-400 mV/dec seems to be off compared to typically reported values.

Response: We highly appreciate this important comment. The Tafel slope of BOC-C and repeated experiments by collecting 4 points were added. The corresponding results and error bars are plotted in Figure R32, which demonstrates that the range of 200-400 mV dec⁻¹ for Tafel slope is relatively accurate.

Based on our DFT calculations, water dissociation and the first electron transfer are involved in rate-limiting process. If the reaction kinetics is fast, the activity will be improved significantly with lower overpotential and smaller Tafel slope will be observed according to Equation R1. That is, the smaller Tafel slope indicates the favorable reaction kinetics (*J. Electrochem. Soc.* **169**, 106505 (2022); *Electrochimica Acta* **357**, 136840 (2020)).

$$\eta = a + b \log j \quad \text{Equation R1}$$

where, η is the overpotential, j is the corresponding current density, a is the Tafel intercept and b is the Tafel slope. The Tafel slope can also be decomposed into

$$b = \frac{-2.3RT}{\alpha nF} \quad \text{Equation R2}$$

where, R is molar gas constant, T is the temperature, α is the transfer coefficient, n is the number of transfer electrons and F is Faraday constant.

Compared with BOC-C, the Tafel slope of BOC-2 is smaller, which implies the faster reaction kinetics. It can be noted that the values of Tafel slope of BOC samples are much larger than 118 mV dec⁻¹, the typical value which indicates that the first electron transfer is involved in rate-limiting process during 2e CO₂RR. As far as we known, there are two major reasons for this large Tafel value as follows:

- (i) The first reason is that the transfer coefficient for metal oxide offsets the empirical value of 0.5 due to the oxide barrier of metal oxide (Modern aspects of electrochemistry. Springer, Boston, MA, 249-300 (2002); *Geochimica ET Cosmochimica Acta* **118**, 56–71 (2013)). As can be seen in equation R2, it can be concluded that the smaller the transfer coefficient, the larger the value of Tafel slope. Thus, the value of Tafel slope for metal oxide with intrinsic insulating layer will be larger than 118 mV dec⁻¹, which is consistent with previous works about metal oxide as summarized in Table R10.
- (ii) If the chemical step precedes the electron transfer or the chemical step is rate-limiting, the Tafel slope will be much larger. According to our DFT calculations and KIE experiments, water dissociation, as the chemical step without potential dependence, is involved in rate-limiting process and precedes the first electron transfer. To reflect the rate-limiting process more comprehensively, the "virtual energetic span" (denoted as δE_v , *ACS Catal.* **8**, 10590-10598 (2018)), which is the difference between turnover frequency determining intermediate (TDI, CO₃* in our work) and turnover frequency determining virtual step (TDTS_v, step IV for BOC),

was introduced as the activity determining term. Besides, the Tafel slope is proportional to the derivative of δE_v with respect to the overpotential ($\partial\delta E_v/\partial U$). As marked in dashed line in Figure R33, the δE_v of BOC decreases with more negative potential applied, indicating the activation energy is affected by applied potential. For BOC with excessive V_O (BOC- V_O -2), the TDI and TDTS_v are CO_3^* and step III, respectively, which manifests that the rate-limiting process is a chemical step without electron transfer. Correspondingly, the value of δE_v for BOC- V_O -2 in DFT calculations is less affected by applied potentials and the Tafel slope of BOC-4 with more V_O in our experiments is much larger than that of other samples.

Changes have been made in the revised manuscript (Page 7, Line 165):

"... Tafel plots were then obtained at sufficient low overpotential to investigate the role of V_O in the kinetics of CO_2RR . As can be seen from Fig. 3a, BOC-2 shows smaller Tafel slope among all catalysts, indicating that the introduction of V_O favors the kinetics of CO_2RR ⁵⁷. It's noteworthy that the value of Tafel slope is much larger than the reported typical value, which indicates that the chemical step precedes the electron transfer or the chemical step is rate-limiting. The detailed investigation of the reaction mechanism will be discussed later. ..."

Figure R32. The Tafel slope of BOC-C, BOC-1, BOC-2, BOC-3 and BOC-4. Due to the overpotential of BOC-C is higher than that of other BOC with V_O samples, the potential range for BOC-C is a little different compared with other samples (corresponding to Fig.3a in the revised manuscript).

Figure R33. Free energy diagram (FED) of BOC and BOC-V₀-2 at different potentials. The arrows plotted in the FED point from the TDI (the horizontal line) to the TDTS_v (marked by red rectangle), their difference (δE_v) is proved to be in direct proportion to activity.^[10]

Table R10. The Tafel slope of metal oxide in CO₂RR reported recently.

Sample	Electrolyte	Tafel slop (mV dec ⁻¹)	Main product in CO ₂ RR	Ref
CuO/carbon black	0.1 M KHCO ₃	246	CO	[1]
Co(TAP)	0.2 M PBS + 0.5 M KHCO ₃	270	CO	[2]
Bi ₂ O ₃	0.5 M KHCO ₃	227	formate	[3]
Bi/CeO _x	0.2 M Na ₂ SO ₄	168	formate	[4]
Bi ₂ O ₃ /BiO ₂	0.5 M KHCO ₃	275	formate	[5]
Bi ₂ Al ₄ O ₉	0.5 M KHCO ₃	429	formate	[5]
BOC/G	0.1 M KHCO ₃	314	formate	[6]
BiO _x /C	0.1 M KHCO ₃	341	formate	[7]
Bi ₂ O ₃ @rGO	0.1 M KHCO ₃	247	formate	[8]
Bi ₂ O ₃ /CNT	0.5 M KHCO ₃	183	formate	[9]

References

- Li, Y. et al. Achieving highly selective electrocatalytic CO₂ reduction by tuning CuO-Sb₂O₃ nanocomposites. *ACS Sustainable Chem. Eng.* **8**, 4948–4954 (2020).

2. Lin, S. et al. Covalent organic frameworks comprising cobalt porphyrins for catalytic CO₂ reduction in water. *Science*, **349**, 1208-1213 (2015).
3. Chen, Z., Mou, K., Wang, X. & Liu, L. Nitrogen - doped graphene quantum dots enhance the activity of Bi₂O₃ nanosheets for electrochemical reduction of CO₂ in a wide negative potential region. *Angew. Chem. Int. Ed.* **57**, 12790 –12794 (2018).
4. Duan, Y. et al. Boosting production of HCOOH from CO₂ electroreduction via Bi/CeO_x. *Angew. Chem. Int. Ed.* **60**, 8798 –8802 (2021).
5. Feng, X. et al. Bi₂O₃/BiO₂ nanoheterojunction for highly efficient electrocatalytic CO₂ reduction to formate. *Nano Lett.* **22**, 1656–1664 (2022).
6. Wang, X. et al. Enhanced electroconversion CO₂-to-formate by oxygen-vacancy-rich ultrasmall Bi-based catalyst over a wide potential window. *ChemCatChem* **14**, e202101873 (2022).
7. Zhao, X. et al. Oxygen vacancies enriched Bi based catalysts for enhancing electrocatalytic CO₂ reduction to formate. *Electrochimica Acta* **367**, 137478 (2021).
8. Yang, X. et al. Partial sulfuration-induced defect and interface tailoring on bismuth oxide for promoting electrocatalytic CO₂ reduction. *J. Mater. Chem. A.* **8**, 2472–2480 (2020).
9. Liu, S. et al. Electronic delocalization of bismuth oxide induced by sulfur doping for efficient CO₂ electroreduction to formate. *ACS Catal.* **11**, 7604–7612 (2021).

Comment 3: KIE analysis

The authors further investigated the KIE by exchanging H₂O with D₂O. Also, they showed that H₂O and not HCO₃⁻ is the major proton source for CO₂RR. While all samples show interesting KIEs and pH dependences, the interpretation of the data seems flawed to this reviewer. The authors argued that a KIE of close to 1 indicates that water dissociation is likely not affecting the catalytic kinetics (which means that H₂O desorption is not rate-limiting), backed up with 1 citation. This reviewer doesn't think that this rationale is correct. The exchange H₂O to D₂O affects a range of properties, which is well known in the literature. Besides the mentioned dissociation effect, H vs D tunnelling probability is the most significant change. The high KIE effect therefore likely reflects proton tunnelling to be rate-limiting, which is also corroborated by the pH-dependence experiments of the authors. Additionally, D₂O has an intrinsically different pD value than H₂O, which should be considered in the discussion. Also, D₂O shows a 20% higher viscosity which likely affects the overall kinetics by affecting mass transport, constitution of the Helmholtz layer etc. All these properties should be included in a discussion before the conclusion of H₂O dissociation to be the driver for the KIE could be made.

Response: We thank the reviewer for this suggestive comment. We agree with the reviewer that the most significant change between D₂O and H₂O is the tunnelling probability. In fact, the theoretical

cornerstone of KIE experiments which is considered to reflect water dissociation ability is the H vs. D tunnelling probability. As schemed in Figure R34, the reaction will happen along the potential energy surface in general (route 1). With the consideration of tunneling probability, the particle will move as marked in red arrow (route 2) due to the probability of matter wave (assuming the wavelength λ). The tunnelling probability is positive correlation with λ while λ_H with small nuclear mass is larger than λ_D according to de Broglie formula. As a result, the dissociation of H₂O requires smaller energy than that of D₂O and the reaction rate constant in H₂O is theoretically much higher than that in D₂O (*J. Phys. Chem. Lett.* **11**, 3724–3730 (2020)). That is, the KIE (the reaction rate ratio between H₂O and D₂O) will be >1 if water dissociation is involved in rate-limiting process.

As for the effect of the difference between pH and pD value, Lan et al. pointed out that the difference in water dissociation constant (K_w) will influence the dissociation probability at equilibrium (F_{eq}) as shown in equation R3 (*J. Phys. Chem. Lett.* **11**, 3724–3730 (2020)).

$$F_{eq} = -k_B T \ln \frac{[OH^-][H_3O^+]}{[H_2O]^2} = -k_B T \ln K_{eq} \quad \text{Equation R3}$$

where $[OH^-]$, $[H_3O^+]$ and $[H_2O]$ are the concentration of the corresponding species within the contact double layer, k_B is the Boltzmann constant, and the T is the temperature. After calculations, it can be concluded that there is no significant difference of F_{eq} in H₂O and D₂O.

To investigate the effect of viscosity in mass transfer, the finite element method based simulation about the diffusions of CO₂ (aq) in H₂O and D₂O was carried out additionally (Figure R35). The diffusion coefficients are $1.9 \times 10^{-9} \text{ m}^2 \text{ s}^{-1}$ in H₂O and $1.65 \times 10^{-9} \text{ m}^2 \text{ s}^{-1}$ in D₂O, respectively. The diffusion thickness of BOC in H₂O is slightly larger than that in D₂O (5.00 μm of H₂O vs. 4.74 μm of D₂O). As a result, the current difference is estimated to be 5%-10% when switching the electrolyte from H₂O to D₂O. Besides, the bulk concentration of CO₂ in D₂O is a little higher than that in H₂O due to the higher solubility (*Biochemistry* **16**, 26 (1977)), which will conversely shrink this diffusion current gap between H₂O and D₂O. While, the smallest KIE value in our work is of 1.19 which is larger than the diffusion current difference induced by mass transfer. In addition, the KIE experiments were carried out with stirred electrolyte in order to mitigate the limit of mass transport. Overall, the reaction rate ratio (k_{H_2O}/k_{D_2O}) is predominately affected by water dissociation due to tunneling effect while the effect of pH and viscosity on the KIE value can be negligible.

Changes have been made in the revised manuscript (Page 8, Line 181):

"... The KIE value is calculated by the ratio of formate production rate in KHCO₃-H₂O and KHCO₃-D₂O. In general, KIE value is >1 if water dissociation is involved in rate-limiting process of formate production due to the proton tunneling effect while the higher KIE value indicates the greater impact of water dissociation in CO₂RR (Supplementary Fig. 28 and detailed discussion can be seen in Supplementary Information)⁶⁰ ..."

Changes have been made in the revised Supplementary Information (Page 2, Line 2):

"...

1. KIE experiments

Herein, the H₂O and D₂O were used in electrolytes to investigate the role of V_o in water dissociation kinetics. Due to the difference in tunneling probability between H and D, the water dissociation barrier can be different, which affects the reaction rate if water dissociation is involved in the rate-limiting step. To be specific, the reaction of water dissociation will proceed along the potential energy surface in general (Supplementary Fig. 28a, route 1). With the consideration of tunneling probability, the particle will move as marked in red arrow (route 2) due to the probability of matter wave (assuming the wavelength λ). The tunnelling probability is positive correlation with λ while λ_H with small nuclear mass is larger than λ_D according to de Broglie formula. As a result, the dissociation of H₂O requires smaller energy than that of D₂O¹⁻². If water is involved in RLS, the KIE (the reaction rate ratio between H₂O and D₂O) will be >1 and the higher KIE value indicates that water dissociation plays a more important role in rate-limiting step. Apart from the H vs. D tunneling probability, the water dissociation is also influenced by water adsorption and OH adsorption. Herein, the water dissociation can be accelerated by improving OH* adsorption and reducing the energetic barrier of water dissociation due to the introduction of V_o. Thus, the difference between the reaction rate in H₂O and D₂O is smaller exhibited as the decreased KIE value.

To investigate the effect of the difference of viscosity between H₂O and D₂O, we have executed a finite element based simulation about the diffusions of CO₂ (aq) in H₂O and D₂O on planar electrode surfaces³⁻⁵. For H₂O, the diffusion coefficient is set to be $1.9 \times 10^{-9} \text{ m}^2 \text{ s}^{-1}$, and the velocity field in convection is set to be $-5 \times 10^{-4} \text{ m s}^{-1}$. As for the case of D₂O, we import the Wilke Chang equation to discuss the deviation in diffusion coefficient with the value of $1.65 \times 10^{-9} \text{ m}^2 \text{ s}^{-1}$ in terms of the viscosity coefficient⁶. The results of the concentration distribution are shown in Supplementary Fig. 28b, which indicates that the diffusion thickness of BOC in H₂O is slightly larger than that in D₂O (5.00 μm of H₂O vs. 4.74 μm of D₂O). As a result, the current difference is estimated to be 5%-10% when switching the electrolyte from H₂O to D₂O. Besides, the bulk concentration of CO₂ in D₂O is a little higher than that in H₂O due to the higher solubility, which will conversely shrink this diffusion current gap between H₂O and D₂O. While, the smallest KIE value in our work is of 1.19 which is larger than the diffusion current difference induced by mass transfer. In addition, the KIE experiments were carried out with stirred electrolyte in order to mitigate the limit of mass transport. Overall, the reaction rate ratio (k_{H_2O}/k_{D_2O}) is predominately affected by water dissociation probability due to tunneling effect and the KIE value can be used to reflect the water dissociation probability.

..."

Figure R34. Schematic diagram of the H-OH cracking in water dissociation (corresponding to Supplementary Figure 28a in the revised Supplementary Information).

Figure R35. The finite element method based on simulation of CO₂ diffusion in a H₂O and b D₂O. c CO₂ concentration against the distance from electrode in H₂O and D₂O (corresponding to Supplementary Figure 28c and 28b in the revised Supplementary Information).

Comment 4: In-situ Raman studies

The authors conducted in situ Raman studies to investigate the fate of the material during catalysis. The overall Raman experiment is well designed. Unfortunately, the Raman spectra are a bit noisy and overall low in intensity, making it hard to unambiguously assign bands and changes. The discussion of

the data is vague and important statements are missing for the reader to understand and follow the rationale. In total, this reviewer believes that the presented in situ Raman spectra and particularly the discussion of the carbonate bands do not conclusively show CO₃ adsorption.

Response: We appreciate the reviewer's valuable comments. The signal to noise ratio in Raman spectra is correlated with acquisition time and accumulation times. To collect the signal in time and reflect the variation more accurately, the Raman spectra were obtained with shorter acquisition time (10 s) and fewer accumulation times (10 times) which can track the whole process during CO₂RR with recognizable signals, even though the signal to noise ratio has to be partially sacrificed. Apart from this, reactor design including the electrolyte and optical window thickness affects the signal intensity as well.

Besides, we are sorry that the mechanism discussed here is not clear as we expected. In the revision version, more comprehensive investigation and explanation of mechanism were carried out as can be seen from the responses to the following detailed comments.

4.1 It would be good to also show the higher frequency region from 1200-1700 cm⁻¹ to monitor the changes ongoing with in the C-O region if possible. In this region, other intermediates, such as Bi-OCO etc., may be observed. The high frequency region holds therefore a lot of information that aids to support the mechanistic model. To rule out such other pathways, the high frequency region may support the mechanistic model.

Response: In light of the reviewer's valuable comment, the in-situ Raman experiments were carried out in a wide range of wavenumbers. As shown in Figure R36, the signal at 1364 cm⁻¹ assigned to the O-C-O symmetric vibration of OCHO* can be observed at -0.78 V (*Adv. Energy Mater.* 2202818 (2022); *ACS Catal.* 9, 9411-9417 (2019)). With more negative potential applied, the intensity of OCHO* increases continuously, indicating that OCHO* is the intermediate in formate production. The observation of OCHO* is one of the strong evidence for supporting the reaction process obtained from theoretical calculations (Figure R37).

Changes have been made in the revised manuscript (Page 12, Line 274):

"... Specially, the signal at 1364 cm⁻¹ assigned to the O-C-O symmetric vibration of OCHO* can be observed at -0.78 V while the intensity of OCHO* increases continuously with more negative potentials applied, indicating that OCHO* is the intermediate in formate formation⁷²⁻⁷³. The observation of OCHO* intermediate provides a strong support for the reaction mechanism proposed in our DFT calculations. ..."

Figure R36. In-situ Raman experiments at different potentials.

Figure R37. The reaction pathway of CO₂RR to formate for BOC based on DFT calculation.

4.2 The authors concluded the stability of the BOC materials from the largely unaltered Raman band structure upon going from OCV to cathodic potentials. This seems to be correct. However, the signal intensity is decreased at cathodic potentials. Therefore, the authors should present the Raman spectra at OCV after the potential series to demonstrate that the original spectrum can be re-obtained.

Response: We thank the reviewer for this valuable suggestion. After a series of potentials applied, the Raman spectra of BOC-2 were collected at OCV again and all these peaks could be kept well without the appearance of metallic Bi signals, demonstrating the stability of BOC-2 under a series of negative

potentials (Figure R38). In addition, it can be noted that the intensity of corresponding signal decreased slightly, which can be attributed to the poor laser tolerance of BOC. In general, the in-situ Raman experiments were carried out by selecting one certain point to monitor the real change during CO₂RR. Thus, the structure of BOC may be destroyed by strong laser after collecting several spectra.

Apart from Raman measurement, the BOC-2 was tested at high current density of 200 mA cm⁻² to investigate the stability. After 15 h, the selectivity of formate can be kept without obvious change in bulk phase (Figure R39), which demonstrates its good stability in CO₂RR.

Changes have been made in the revised manuscript (Page 12, Line 272):

"... Then, the Raman spectra of BOC-2 were collected at different potentials (from -0.58 V to -1.08 V) and all these peaks could be kept well without the appearance of metallic Bi signals, demonstrating the stability of BOC-2 under a series of negative potentials (Fig. 5b). Specially, the signal at 1364 cm⁻¹ assigned to the O-C-O symmetric vibration of OCHO* can be observed at -0.78 V while the intensity of OCHO* increases continuously with more negative potentials, indicating that OCHO* is the intermediate in formate formation⁷³⁻⁷⁴. The observation of OCHO* intermediate provides a strong support for the reaction mechanism proposed in our DFT calculations. ..."

Figure R38. In-situ Raman experiments at different potentials (corresponding to Fig. 5b in the revised manuscript).

Figure R39. a The stability test of BOC-2 at 200 mA cm^{-2} . b XRD patterns of BOC-2 before and after stability test, where there is no characteristic peak of metallic bismuth phase (corresponding to Supplementary Figure 18 in the revised Supplementary Information).

4.3 The interpretation of the $^{12}\text{CO}_2$ and $^{13}\text{CO}_2$ experimental Raman spectra is not fully clear to this reviewer. On page 11 line 250, the authors state that adding $^{12}\text{CO}_2$ results in an $^{12}\text{CO}_3$ peak. In Figure 5c, the spectral region is marked in blue. The black spectrum shows the intrinsic BOC exhibiting the band at 1067 cm^{-1} originating from CO_3 groups of the BOC. But when $^{12}\text{CO}_2$ is added only one band is visible (OCV pale blue spectrum) which is low frequency shifted. If the lower frequency band is the $^{12}\text{CO}_3$ species, where is the band for the intrinsic CO_3 of the BOC? Also, it should be clarified how much time was waited between the spectra in $^{12}\text{CO}_2$ and $^{13}\text{CO}_2$. The authors should also clarify that the exchange does not occur at potentials lower than the onset potentials.

Response: We thank the reviewer for this insightful question. The mechanism of CO_3 involvement in the formation of formate has been proposed previously but the convincing experimental capture of CO_3

species is still in absence (*ACS Omega* **2**, 2561–2567 (2017); *ACS Catal.* **11**, 4988–5003 (2021)). Inspired by these excellent works, we try to verify the CO₃ involvement through theoretical calculations and experimental design.

From the perspective of theoretical investigation on this mechanism, DFT calculations of the whole CO₂RR process were carried out at first. During electroreduction, the intrinsic CO₃²⁻ in BOC interacted with electrolyte will spontaneously undergo charge rearrangement and be in-situ transformed into CO₃* species with an unpaired electron (Figure R40). The CO₃* is more active and participates in CO₂RR to produce formate as the surface species. After that, one oxygen site was left and the input CO₂ can be easily adsorbed on oxygen site to form CO₃* (due to the downhill process, Figure R41) which proceeds the next cycle (Figure R42). That is, DFT calculations indicate that CO₃* is the surface species participating in CO₂RR.

To verify this mechanism in experiments, EPR measurements were carried out benefitting from the presence of unpaired electron in CO₃* which can be captured by trapping agent. As shown in Figure R43, the typical sextet appears during CO₂ electroreduction, which implies the presence of CO₃*. It can be noted that the CO₃* is in absence without CO₂, which can be ascribed to that minor CO₃* existed in electrode will be consumed rapidly and hard to be captured by trapping agent. During CO₂ electroreduction, the CO₃* is more available due to the continues input of CO₂ and the CO₃* can be more easily to be captured as a result.

Then, in situ Raman measurements were further implemented to probe the related species directly along the reaction route. Prior to characterization, the Raman vibrational features of CO₃* was calculated and added in revised version as shown in Figure R44. The two major vibration modes were selected from more than 100 potential modes of vibration while the position of C-O vibration in CO₃* blueshifts mildly relative to that in CO₃²⁻, which is consistent with the reported experimental result (1063 cm⁻¹ of CO₃* and 1065 cm⁻¹ of CO₃²⁻, *Laser Chem.* **19**, 311-316 (1999)). The slight gap on specific wavenumbers between calculations and experiments may be induced by the simplify of real condition in calculations. For Raman measurement, the signal of C-O vibration of BOC at OCV is broaden and shifts towards low wavenumbers (1067 cm⁻¹) as compared with intrinsic signal of carbonate (1069 cm⁻¹) in BOC sample, which indicates the transformation of CO₃* from intrinsic CO₃²⁻ at least occurs under OCV (Figure R45). Apart from that, the signal at 1067 cm⁻¹ is almost disappear accompanied with the appearance of ¹³CO₃* during ¹³CO₂ electroreduction (Figure R46). So, it is reasonable to conclude that the signal at 1067 cm⁻¹ is at least mainly induced by CO₃* based on above analysis even though the contribution of intrinsic CO₃²⁻ cannot be totally ruled out.

Finally, in order to verify the cycle of CO₃*, isotopic labeling experiments was designed. Ar was purged into the original electrolyte for 15 min to exclude the presence of ¹²CO₂ (Figure R46). Next, ¹³CO₂ was purged into the electrolyte for additional 15 min in order to achieve saturation. At -0.18 V

which is far away from the onset potential of formate, only the signal of $^{12}\text{CO}_3^*$ can be observed. The absence of OCHO^* proves that there is no apparent formate formation at this potential. When a more negative potential of -0.88 V was applied, the signals of $^{13}\text{CO}_3^*$ and $^{12}\text{CO}_3^*$ were observed simultaneously accompanied by the appearance of $\text{O}^{13}\text{CHO}^*$ and $\text{O}^{12}\text{CHO}^*$. After 5 min electroreduction, there were just the signals of $^{13}\text{CO}_3^*$ and $\text{O}^{13}\text{CHO}^*$. The overall process implies that CO_3^* indeed participates in formate formation as the surface species.

With the combination of theoretical calculations and experiments including EPR measurements and in-situ Raman characterization combined with isotopic labeling, it can be clarified that CO_3^{2-} in BOC can be in-situ activated as CO_3^* during CO_2RR which participates in the formate production as surface species.

Changes have been made in the revised manuscript (Page 11, Line 254):

"... For CO_2 adsorption and forming surface species, DFT calculations show that intrinsic CO_3^{2-} in BOC will spontaneously undergo charge rearrangement and be in-situ transformed into CO_3^* which participates in the sequent formate production with a O^* site left (Supplementary Fig. 36). Then, CO_2 prefers to adsorb on this oxygen site to form CO_3^* species again than adsorb on bismuth. The charge distribution of CO_3^* based on Bader charge analysis indicates the CO_3^* is in the form of radical with an unpaired electron (Supplementary Fig. 37). To clarify the presence of CO_3^* , EPR measurements were first performed with the trapping agent of 5,5-dimethyl-1-pyrroline-N-oxide (DMPO). In the absence of CO_2 , there is just the nonet ascribed to H^* radicals (hyperfine splitting constants, $A_N = 1.65\text{ mT}$, $A_H = 2.25\text{ mT}$) which are originated from water dissociation (Fig. 5a)⁶⁸. With the feed of CO_2 , the characteristic sextet of carbon radicals can be found ($A_N = 1.59\text{ mT}$, $A_H = 2.28\text{ mT}$), which agrees well with theoretical analysis⁶⁹. These observations imply the presence of CO_3^* during CO_2RR process.

Then, in-situ Raman measurements were implemented supplementarily to probe the related species directly along the reaction route. For the intrinsic Raman spectrum of BOC, the bands at 155 cm^{-1} and 367 cm^{-1} are the characteristic vibrational modes of $\text{Bi}=\text{O}$ bond lattice in $[\text{Bi}_2\text{O}_2]^{2+}$. The peak centered at 1069 cm^{-1} is attributed to the typical C-O stretching of carbonate in $\text{Bi}_2\text{O}_2\text{CO}_3$ (Supplementary Fig. 38)⁷⁰. At open circuit voltage (OCV), this signal blueshifts to the center of 1067 cm^{-1} , which indicates the transformation of CO_3^* from intrinsic CO_3^{2-} at least occurs under OCV in consideration of DFT calculations and EPR results⁷¹. Similar results are obtained in the electrolyte of K_2HPO_4 , which excludes the interference of HCO_3^- and CO_3^{2-} in electrolyte. Then, the Raman spectra of BOC-2 were collected at different potentials (from -0.58 V to -1.08 V) and all these peaks could be kept well without the appearance of metallic Bi signals, demonstrating the stability of BOC-2 under a series of negative potentials (Fig. 5b). Specially, the signal at 1364 cm^{-1} assigned to the O-C-O symmetric vibration of OCHO^* can be observed at -0.78 V while the intensity of OCHO^* increases continuously with more negative potentials, indicating that OCHO^* is the intermediate in formate formation⁷²⁻⁷³. The

observation of OCHO* intermediate provides a strong support for the reaction mechanism proposed in our DFT calculations.

To verify the cycle of CO₃* in CO₂RR, three steps with isotopic labelling were designed by using ¹²CO₂ and ¹³CO₂ as the carbon source interchangeably. First, ¹²CO₂ was introduced into the reactor, and there is an obvious peak of ¹²CO₃* accompanied with Bi=O and ¹²OCHO* vibration at -0.88 V (Fig. 5c and Supplementary Fig. 39). Subsequent to Ar purging to remove the residual dissolved ¹²CO₂ in reactor, ¹³CO₂ was introduced into the system for 15 min to achieve saturation. It can be noted that a new and wide peak at 1012 cm⁻¹ appears which indicates the presence of ¹³CO₃* based on the isotopic effect⁷⁴. The slight difference of peak position between experiments and theoretical calculations may be derived from solvent effect and applied potential⁷⁵. Moreover, the intensity of ¹³CO₃* increases accompanied with the intensity decrease of ¹²CO₃* as time goes on, indicating that the initial ¹²CO₃* in the first step is gradually consumed and replaced by ¹³CO₃*. It's noteworthy that there was no similar exchange at OCV and -0.18 V far away from the onset potential of formate production, which suggests that the observed exchange is indeed resulted from the involvement of CO₃* in CO₂RR. In the final step, ¹²CO₂ was fed into the system again and the intensity of ¹²CO₃* increases impressively with the gradual disappear of ¹³CO₃*, which manifests clearly that CO₃* is involved in CO₂RR as the key surface species. ..."

Figure R40. Bader charge distribution of surface species of CO₃*. The purple, red, brown, and white balls represent bismuth (Bi), oxygen (O), carbon (C) and hydrogen (H), respectively.

Figure R41. The free energy of CO_3^* formation which is exothermic.

Figure R42. The schematic plot of catalytic cycle in CO_2RR based on DFT analysis. Among this, brown balls represent bismuth (Bi), green balls represent oxygen (O), blue balls represent carbon (C) and white

balls represent hydrogen (H), respectively (corresponding to Supplementary Fig.36 in the revised Supplementary Information).

Figure R43. EPR trapping of CO_3^* species.

Figure R44. Computed Raman vibrational features for CO_3^{2-} and CO_3^* . a The major signals of CO_3^{2-} and CO_3^* in Raman spectra. b The corresponding vibrational modes (corresponding to Supplementary Fig.38b and c in the revised Supplementary Information).

Figure R45. The Raman spectra of intrinsic BOC-2 and BOC-2 at OCV in KHCO_3 and K_2HPO_4 . The intrinsic spectrum is obtained at BOC-2 sample without electricity (corresponding to Supplementary Fig.38a in the revised Supplementary Information).

Figure R46. The in-situ Raman spectra of BOC with $^{12}\text{CO}_2$ and $^{13}\text{CO}_2$ as carbon source interchangeably (corresponding to Supplementary Fig.39 in the revised Supplementary Information).

4.4 Same question as in 4.2 arises when $^{13}\text{CO}_2$ is introduced. The intrinsic CO_3 band of BOC is missing. The rising lower frequency band at 1012 cm^{-1} may originate from $^{13}\text{CO}_3$ formed. How can the authors be sure that this reflects carbonate formed at the material interface/ at the material. Purging CO_2 into KHCO_3 0.5 M should change the bicarbonate/carbonate equilibrium and hence affects the carbonate band. Fig 5b shows the Raman spectrum of the electrolyte lacking any carbonate/bicarbonate bands, which is unusual given the high concentration of 0.5 M. To demonstrate that the 1067 cm^{-1} bands are coming from solely from the material interaction, the authors should show the Raman spectra in Ar-saturated electrolyte under potential in 0.5 M KHCO_3 and in KPHO_4 (same potentials as shown in Fig 5b). The reason is that it is not clear where the carbonate bands are coming from, i.e., electrolyte saturation or material or both, and why the material's intrinsic carbonate is not observed.

Response: We thank the reviewer for this insightful question and valuable suggestions. The spectra of electrolyte were collected again during the in-situ Raman experiments, and there was no signal can be discernible, which demonstrates the signal of CO_3^* is exactly from electrode rather than electrolyte (Figure R47). The absence of electrolyte signal consists with previous works (*Nat. Commun.* **13**, 2039 (2022); *Nat. Commun.* **11**, 3415 (2020)), even though the HCO_3^- can be detected by some works. As far as we considered, the signals of HCO_3^- and CO_3^{2-} may be too weak to be observed due to the limitation of in-situ Raman cell where the thickness of electrolyte and optical window can influence the intensity of signals to a large extent.

In light of the reviewer's suggestion, the in-situ Raman experiments were carried out specifically in CO_2 -saturated K_2HPO_4 to further exclude the influence of HCO_3^- . As shown in Figure R48, the CO_3^* can be well kept accompanied by the signal of OCHO^* , which indicates the presence of CO_3^* rather than the interference of electrolyte. After replacing CO_2 by Ar, the signal of CO_3^* disappears, which indicates that CO_3^* participates in CO_2RR as the surface species. For the ascription of Raman peak of 1067 cm^{-1} , as discussed in detail in the response of 4.3, this signal is mainly from CO_3^* which is in-situ transformed from intrinsic CO_3^{2-} in BOC based on our DFT calculations, EPR and Raman characterizations.

Changes have been made in the revised manuscript (Page 13, Line 297):

"... Besides, the signal of CO_3^* disappears when CO_2 was replaced by Ar, which suggests the involvement of CO_3^* as well. (Supplementary Fig. 40). ..."

Figure R47. The Raman spectra of electrolyte at different conditions and there is no characteristic of signals of HCO_3^- and CO_3^{2-}

Figure R48. The in-situ Raman spectra of BOC-2 with CO₂-saturated 0.5 M KHCO₃, Ar-saturated 0.5 M KHCO₃, CO₂-saturated 0.5 M K₂HPO₄ and Ar-saturated 0.5 M K₂HPO₄ solutions as electrolyte,

respectively. The green areas represent the characteristic signals of CO_3^* and the red area is the signal of the OCHO^* originated from CO_2RR . After replacing CO_2 by Ar, the intensity of CO_3^* decreases notably with the disappear of OCHO^* signal, indicating that the involvement of CO_3^* in formate production (corresponding to Supplementary Fig.40 in the revised Supplementary Information).

4.5 This reviewer appreciates the fact that the comparison to BOC-3 and BOC-C has been made. However, the interpretation of Fig 5e is not clear. BOC-C seems to exhibit partial exchange. The authors should discuss this and what this means in the context of their hypothesis.

Response: We thank the reviewer for this comment. As discussed above, the charge of CO_3^{2-} can be reorganized and CO_3^{2-} is in-situ transformed into CO_3^* which participates in CO_2RR as surface species. The partial exchange in BOC-C indicates that there are some CO_3^{2-} fail to be transformed into the active CO_3^* , which may be one of the reasons for the poor activity of BOC-C. In addition, it can be noted that the intensity of residual signal at 1067 cm^{-1} during $^{13}\text{CO}_2\text{RR}$ (Line 4) is much lower than that during $^{12}\text{CO}_2\text{RR}$ (Line 2), which implies that CO_3^* is dominate in the contribution of 1067 cm^{-1} signal (Figure R49).

Changes have been made in the revised manuscript (Page 13, Line 295):

"... The partial exchange in BOC-C after $^{13}\text{CO}_2\text{RR}$ indicates that there are some CO_3^{2-} fail to be transformed into the active CO_3^* , which may be one of the reasons for the poor activity of BOC-C. ..."

Figure R49. In situ Raman spectroscopy with $^{12}\text{CO}_2$ and $^{13}\text{CO}_2$ as carbon source for BOC-C.

Reviewer #3 (Remarks to the Author):

*The authors investigate the effect of oxygen vacancies in $\text{Bi}_2\text{O}_2\text{CO}_3$ (BOC) on enhancing the CO_2 reduction reaction to formate by boosting the water dissociation. The experimental results and theoretical calculation demonstrate that oxygen vacancies stabilize the hydroxyl group and drive efficient water dissociation. Consequently, the energy barrier for *OCHO intermediate is reduced on oxygen sites of BOC. This work is done well and clearly shows the evidence for their claim. However, the reviewer can find one major inconsistency in BOC electrocatalysts compared to the previous publications and minor issues that should be thoroughly addressed via major revision.*

Response: We thank the reviewer very much for the positive comments and valuable suggestions, which help us improve the quality of our work further. We have thought of your comments carefully and done supplementary experiments to address these concerns. Please see our point-by-point responses as follows.

1) In Figure S1, there are several XRD peaks. BOC nanosheets appear to be polycrystalline rather than single crystalline.

Response: We thank the reviewer for this comment. As can be seen from Figure R50a, the BOC are consisted of stacked nanosheet with different orientations, which display the polycrystalline feature indicated by the diffraction rings in Figure R50b. This result is consistent with XRD results. For individual nanosheet (Figure R51), the selected area electron diffraction shows an array of bright spots, which suggests the single crystalline property. Thus, it can be concluded that the BOC with stacked nanosheets display polycrystalline feature that stems from the individual nanosheet with single crystalline property, which is in good agreement with previous works (*ChemPhysChem* **11**, 2167-2173 (2010); *J. Alloys and Compd.* **769**, 301e310 (2018); *Cryst. Growth Des.* **15**, 534-537 (2015)).

Changes have been made in the revised manuscript (Page 3, Line 88):

"... A lattice distance of 0.275 nm corresponding to (110) plane is clearly discerned in high-resolution TEM (HRTEM) images (Fig. 1c), while the selected area electron diffraction (SAED) patterns display the BOC nanosheets are consisted of individual single crystalline nanosheet (inset in Fig. 1c and Supplementary Fig. 3d). ..."

Figure R50. a TEM image and b the corresponding selected area electron diffraction for BOC-2 (corresponding to Supplementary Fig.3c and 3d in the revised Supplementary Information).

Figure R51. HRTEM image of single nanosheet and the inset is the corresponding selected area electron diffraction.

2) The reviewer cannot find the defect in Figures 1d and S5. Please explain how to identify one.

Response: We thank the reviewer for this comment. As shown in dark field TEM image (Figure R52), the bright spots represent Bi atom and some small pits can be observed as circled by white dashed line. Within these pits, lattice disorders can be observed, exhibiting some atoms deviate from their original positions (Figure R52b), which indicates the presence of defects (*J. Am. Chem. Soc.* **138**, 8928–89352 (2016); *Nat. Commun.* **10**, 788 (2019)).

Figure R52. High-angle annular dark-field TEM images of BOC-2 where the scale bar is 2 nm in a and 0.5 nm in b, respectively (corresponding to Fig. 1e in the revised manuscript).

3) In Figure 1g, the y-axis values show less than 3% difference in intensity that might be within the experimental error. Also, why the presence of oxygen vacancy shows a larger intensity in the y-axis than that of oxygen sites?

Response: Thanks for this valuable comment. With the model of bright-field STEM, the atoms are dark with the bright background. If there is no atom, the intensity would be higher due to the bright background. That's why the intensity of oxygen vacancy is relative higher than that of oxygen site. As shown in Figure R53, the relatively large black hole is the bismuth atom while the small hole can be assigned to oxygen atom due to the difference in atomic number. To eliminate the effect of error, three areas were selected as shown in Figure R54 and the intensity of oxygen atom is similar, with a maximum difference of 2000 a.u. (section 2). In contrast, the difference between oxygen vacancy and oxygen atom is more than 6000 a.u., much larger than the error of oxygen atom intensity (section 1 and section 3). Therefore, the identification of V_O from STEM is relatively reliable, which are supported by previous works as well (*Appl. Catal. B: Environ.* **306**, 121100 (2022); *Nat. Commun.* **11**, 1664 (2020); *Nat. Commun.* **8**, 104 (2017)). Apart from STEM, synthetic characterizations including EPR, XPS and XAS are carried out, which together demonstrate the presence of oxygen vacancy (Figure R55).

Figure R53. ABF-STEM bright filed image of BOC-2, where the pink and blue sites represent oxygen and bismuth atoms, respectively. The right plot is the corresponding intensity profile as marked by the orange arrow. Among this, the trough with lower intensity indicates the presence of Bi or O site at this position.

Figure R54. ABF-STEM bright filed image and the corresponding intensity profile of BOC-2. The blue balls represent bismuth while the purple balls represent oxygen. Three selections were selected randomly and section 1 and 3 contain the oxygen vacancy and oxygen sites while there are only oxygen sites in section 2 (corresponding to Supplementary Fig. 6 in the revised Supplementary Information).

Figure R55. The identification of V_O . a EPR spectra of the all samples, where the g value of 2.002 is the characteristic g factor of V_O . b Fourier transform of Bi L_3 edge EXAFS data recorded at R space. c XPS spectra of BOC-C, BOC-1, BOC-2, BOC-3 and BOC-4 on Bi 4f, where the green and pink area represent Bi^{3+} and $Bi^{(3-x)+}$, respectively.

4) Why does controlling the water:ethylene glycol ratio change the oxygen vacancies concentration in BOC? A brief description should be at least provided.

Response: We thank the reviewer for this comment. In general, BOC with V_O was synthesized in the mixture solvent of ethylene glycol and water (*Appl. Surf. Sci.* **495**, 143561 (2019); *Inorg. Chem. Front.* **7**, 2969 (2020); *J. Colloid and Interface Sci.* **609**, 320–329 (2022)). According to previous work related $Bi_2O_2MoO_2$ (BMO) synthesis (*J. Hazard. Mater.* **364**, 691-699 (2019)), the hydrogen bond between solvent molecular and BMO is in favor of the breakage of Bi-O bond and increasing the V_O concentration. Guided by this conclusion, the ratio of water in the mixed solvent was increased in our work to strength the hydrogen bond with the purpose of preparing BOC samples with different V_O

concentrations.

Changes have been made in the revised manuscript (Page 3, Line 81):

"... V_O engineering was achieved by tuning the proportion of water in the mixed electrolyte with ethylene glycol, and a series of samples denoted as BOC-1, BOC-2, BOC-3 and BOC-4 respectively, were obtained⁴⁶. The tetragonal structure of $\text{Bi}_2\text{O}_2\text{CO}_3$ is clearly identified from X-ray ..."

(Page 15, Line 341):

"... The electrode was removed and cleaned with deionized water subsequently. Promoting the breakage of Bi-O bond through varying solvent is in favor of increasing the V_O concentration. Here, by varying the volume ratio of ethylene glycol and water in electrolyte, BOC-1 (pure ethylene glycol), BOC-2 (3:1), BOC-3 (1:1) and BOC-4 (pure water) were obtained, respectively. ..."

Reference:

46. Xu, X. et al. Oxygen vacancy boosted photocatalytic decomposition of ciprofloxacin over Bi_2MoO_6 : oxygen vacancy engineering, biotoxicity evaluation and mechanism study. *J. Hazard. Mater.* **364**, 691-699 (2019).

5) If the water dissociation is promoted, hydrogen evolution reaction (HER) might also be enhanced due to the increase in the number of protons near the surface. So, please discuss why the competing HER is not enhanced but only CO_2 reduction is enhanced.

Response: We thank the reviewer for this insightful comment. We agree with the reviewer's opinion that the promoted water dissociation is also beneficial for HER. The j_{hydrogen} of BOC with V_O is slightly larger than that of BOC-C (Figure R56). However, the BOC-4 with more V_O and sluggish water dissociation kinetics shows larger j_{hydrogen} compared with BOC-2 with accelerated water kinetics, which indicates that the production rate of hydrogen is not merely restricted by water dissociation. In fact, the kinetics of HER is predominately restricted by H^* adsorption as demonstrated by previous works (*J. Electrochem. Soc.* **152**, 23-26 (2005); *Phys. Chem. Chem. Phys.* **18**, 23864-23871 (2016)).

In order to get further insight into the H^* adsorption, DFT calculation was added. As can be seen in Figure R57, the energy of H^* adsorption on BOC is much larger than CO_2 adsorption, suggesting the weak H^* adsorption of BOC and CO_2RR will occur preferentially compared with HER. Therefore, the HER performance of BOC with V_O is relatively poor compared with that of CO_2RR , even though beneficial from the accelerated water dissociation. On the other hand, the formate production rate increases sharply by promoting water dissociation because the formate production is mainly limited by water dissociation as discussed in our work. Thus, it's an effective method to promote CO_2RR through promoting water dissociation although concomitant HER performance is improved slightly.

Changes have been made in the revised manuscript (Page 11, Line 231):

"...As can be deduced from Fig. 4e, the energy variation of the step III is mainly from $\Delta\mu_{\text{OH-w}}$, which suggests that the formate production rate is dominantly restricted by the sluggish water dissociation. While the introduction of V_{O} accelerates the kinetics of water dissociation by notably reducing $\Delta\mu_{\text{OH-w}}$, which largely decreases the energy barrier in the formation of OCHO^* . These results are in good agreement with KIE experiments discussed above. On the other hand, the accelerated water dissociation kinetics has minor effects on HER due to the poor H^* adsorption on BOC (Supplementary Fig. 34)..."

Figure R56. The j_{hydrogen} for BOC-C, BOC-1, BOC-2, BOC-3 and BOC-4 at different negative potentials.

Figure R57. a DFT calculations about H^* adsorption, CO_2 adsorption on BOC with V_{O} and H^* on Pt. b the basic model of H^* on BOC with V_{O} . Colours in the model: purple balls are bismuth (Bi); red balls are oxygen (O); gray balls are carbon (C) and blue balls are added hydrogen (H^*) corresponding to

Supplementary Fig. 34 in the revised Supplementary Information).

6) Authors claim that “there is no obvious change in phase structure, morphology and valance state, which demonstrates the stability of BOC-2 (Supplementary Fig. 15).” However, this is in great contradiction with the recently reported BOC catalyst (<https://doi.org/10.1002/eem2.12490>). Based on the Pourbaix diagram of BOC, it should be reduced to Bi under cathodic bias ([https://doi.org/10.1016/S0378-7753\(00\)00640-6](https://doi.org/10.1016/S0378-7753(00)00640-6)). Authors should clearly state the reason for the discrepancy by citing the above references.

Response: We thank the reviewer for this very insightful and valuable comment. To check the stability of BOC-2, several experiments were specifically carried out as follows. First, the CO₂RR performance of BOC-2 was tested at large current density of 200 mA cm⁻² for 15 h, and the FE_{formate} and structure of BOC can be kept well (Figure R58). Second, the in-situ Raman measurements were conducted for BOC-2 at a series of negative potential to monitor the stability of BOC-2 on surface. As can be seen from Figure R59, the characteristic peaks of BOC are retained during the whole process. Thus, the stability of BOC-2 can be demonstrated based on the experimental results above. Besides, the good stability of BOC is also reported by previous work (*Cell Rep. Phys. Sci.* **2**, 100353 (2021)).

According to Pourbaix diagram of BOC (*J. Power Sources* **95**, 108-118 (2001)), Bi₂O₃ will be transformed into metallic Bi at negative potentials thermodynamically. On the other hand, Bi₂O₂CO₃ is verified to be stable in Pourbaix diagram pointed out by Lin. et al. (*Angew. Chem. Int. Ed.* e202214959 (2022)). As far as we known, there are three possible factors which hamper the reduction of BOC:(1) the high local pH resulted from the rapid consumption of protons will lead to the reduced Bi more easily oxidized (Bi + 3OH⁻ → Bi(OH)₃), which has been discussed in previous works (*Cell Rep. Phys. Sci.* **2**, 100353 (2021); *Nat. Commun.* **12**, 5223 (2021)). (2) BOC can be spontaneously transformed from Bi in KHCO₃ solutions (*Energy Environ. Mater.* e12490 (2022)). (3) Based on DFT calculations, the formation of BOC through CO₂ adsorption on Bi site is a downhill process, which indicates BOC can be stable with the presence of CO₂ (Figure R60). That is, the Bi³⁺ tends to be reduced to metallic Bi in thermodynamics at negative potentials while Bi⁰ is easily to be oxidized to be BOC during CO₂RR due to high local pH and accessible CO₂ adsorption.

Changes have been made in the revised manuscript (Page 7. Line 146):

"... To relieve the limitation on mass transfer and pursue commercial current density requirements, BOC-2 was further integrated into gas diffusion electrode (GDE) and evaluated in flow-cell system⁵³.

The j_{formate} of BOC-2 is 1.6 times as high as that of BOC-C at -1.68 V while the turnover frequency can be up to 0.72 s⁻¹ at 200 mA cm⁻², which demonstrates the activity enhancement through introducing V_O

can also be achieved in flow-cell system (Supplementary Fig. 17 and Supplementary Table 4). In addition, a stable FE_{formate} of 80% at large current density of 200 mA cm^{-2} can be observed during 15 h, which indicates the potential for practical application (Supplementary Fig. 18). Even though Bi^{3+} could be reduced at negative potentials in thermodynamics, the good stability of BOC can be explained by the spontaneous CO_2 adsorption and high local pH where the oxide state of Bi is stable^{29, 33, 54-55}. Moreover, $^{13}\text{CO}_2$ labeling experiment was carried out and the proton doublet resulting from H- ^{13}C coupling and $\text{H}^{13}\text{COO}^-$ is observed in ^1H NMR and ^{13}C NMR, respectively (Supplementary Fig. 19), demonstrating that the produced formate derives from CO_2 ⁵⁶."

Figure R58. a The stability test of BOC-2 at 200 mA cm^{-2} . b XRD patterns of BOC-2 before and after stability test, where there is no characteristic peak of metallic bismuth phase (corresponding to Supplementary Fig.18 in the revised Supplementary Information).

Figure R59. The in-situ Raman spectra of BOC-2 at different potentials. The gray and blue areas are the characteristic peaks of BOC (corresponding to Fig.5b in the revised manuscript).

Figure R60. The free energy of CO_2 adsorption which is exothermic.

7) The reviewer also wants to know and recommend explaining why H_2O showed a higher formate production rate compared to D_2O in Figures 3b and 3c.

Response: We thank the reviewer for this valuable comment. Due to the difference in tunneling

probability between H and D, the water dissociation barrier can be different, which affects the reaction rate if water dissociation is involved in the rate-limiting step. To be specific, the reaction of water dissociation will proceed along the potential energy surface in general (Figure R61, route 1). With the consideration of tunneling probability, the particle will move as marked in red arrow (route 2) due to the probability of matter wave (assuming the wavelength λ). The tunnelling probability is positive correlation with λ while λ_H with small nuclear mass is larger than λ_D according to *de Broglie* formula. As a result, the dissociation of H₂O requires smaller energy than that of D₂O and the reaction rate constant in H₂O is theoretically much higher than that in D₂O (*J. Phys. Chem. Lett.* **11**, 3724–3730 (2020)). Therefore, the production rate of formate is higher in H₂O compared with that in D₂O due to the lower energetic barrier of water dissociation mainly induced by good tunneling probability of H.

Changes have been made in the revised Supplementary Information (Page 2, Line 2):

"... **I. KIE experiments**

Herein, the H₂O and D₂O were used in electrolytes to investigate the role of V_O in water dissociation kinetics. Due to the difference in tunneling probability between H and D, the water dissociation barrier can be different, which affects the reaction rate if water dissociation is involved in the rate-limiting step. To be specific, the reaction of water dissociation will proceed along the potential energy surface in general (Supplementary Fig. 28a, route 1). With the consideration of tunneling probability, the particle will move as marked in red arrow (route 2) due to the probability of matter wave (assuming the wavelength λ). The tunnelling probability is positive correlation with λ while λ_H with small nuclear mass is larger than λ_D according to *de Broglie* formula. As a result, the dissociation of H₂O requires smaller energy than that of D₂O¹⁻². If water is involved in RLS, the KIE (the reaction rate ratio between H₂O and D₂O) will be >1 and the higher KIE value indicates that water dissociation plays a more important role in rate-limiting step. Apart from the H vs. D tunneling probability, the water dissociation is also influenced by water adsorption and OH adsorption. Herein, the water dissociation can be accelerated by improving OH* adsorption and reducing the energetic barrier of water dissociation due to the introduction of V_O. Thus, the difference between the reaction rate in H₂O and D₂O is smaller exhibited as the decreased KIE value.

To investigate the effect of the difference of viscosity between H₂O and D₂O, we have executed a finite element based simulation about the diffusions of CO₂ (aq) in H₂O and D₂O on planar electrode surfaces³⁻⁵. For H₂O, the diffusion coefficient is set to be $1.9 \times 10^{-9} \text{ m}^2 \text{ s}^{-1}$, and the velocity field in convection is set to be $-5 \times 10^{-4} \text{ m s}^{-1}$. As for the case of D₂O, we import the Wilke Chang equation to discuss the deviation in diffusion coefficient with the value of $1.65 \times 10^{-9} \text{ m}^2 \text{ s}^{-1}$ in terms of the viscosity coefficient⁶. The results of the concentration distribution are shown in Supplementary Fig. 28b, which indicates that the diffusion thickness of BOC in H₂O is slightly larger than that in D₂O (5.00 μm of H₂O vs. 4.74 μm of D₂O). As a result, the current difference is estimated to be 5%-10% when switching the

electrolyte from H₂O to D₂O. Besides, the bulk concentration of CO₂ in D₂O is a little higher than that in H₂O due to the higher solubility, which will conversely shrink this diffusion current gap between H₂O and D₂O. While, the smallest KIE value in our work is of 1.19 which is larger than the diffusion current difference induced by mass transfer. In addition, the KIE experiments were carried out with stirred electrolyte in order to mitigate the limit of mass transport. Overall, the reaction rate ratio ($k_{\text{H}_2\text{O}}/k_{\text{D}_2\text{O}}$) is predominately affected by water dissociation probability due to tunneling effect and the KIE value can be used to reflect the water dissociation probability. ..."

Figure R61. Schematic diagram of the H-OH cracking in water dissociation. The route 1 is general in reaction while the route 2 is induced by the presence of H tunneling probability (corresponding to Supplementary Fig.28a in the revised Supplementary Information).

REVIEWERS' COMMENTS

Reviewer #1 (Remarks to the Author):

The authors have addressed the concerns of the reviewer and I recommend publication of the article in its revised form.

Reviewer #2 (Remarks to the Author):

This reviewer thanks the authors for their detailed responses to raised comments. This reviewer sees all his comments addressed. The modified manuscript has significantly improved with respect to level of scientific scrutiny and overall quality.

The experimental data presentation has significantly improved and so has the data interpretation.

In total, experimental data, DFT calculations, and thorough discussion of the results yield a coherent study on the role of water dissociation in CO₂RR for this type of electrocatalysts.

Reviewer #3 (Remarks to the Author):

The authors have clearly explained all the questions except for one.

The article cited by the author on the stability of BOC (Angew. Chem. Int. Ed. e202214959 (2022)) also describes the partial reduction of BOC to Bi/BOC binary compound under cathodic reaction conditions. The author mentioned that (1) the reduced Bi could be oxidized to Bi(OH)₃ due to the local pH effect and (2) the reduced Bi can be oxidized to BOC with the presence of CO₂. I also agree that these side reactions can occur during the CO₂ reduction reaction. However, if these reactions happen, the total faradaic efficiency should be lower than 100% and more significantly, these repeated redox reactions will result in the reconstruction of crystal structure and morphology. The single crystalline nature of BOC is very likely altered from its original form (not detectable by in-situ Raman or XRD measurements).

Since the key point of the manuscript, that water dissociation becomes the RDS of the CO₂ reduction reaction, is well explained, this manuscript in its current form can be accepted. However, for the sake of completeness, I strongly suggest confirming the change in the crystal structure after the reaction by STEM measurement.

Reviewer #1 (Remarks to the Author):

The authors have addressed the concerns of the reviewer and I recommend publication of the article in its revised form.

Response: We thank the reviewer for the positive comments and recommendation on the publication of this work.

Reviewer #2 (Remarks to the Author):

This reviewer thanks the authors for their detailed responses to raised comments. This reviewer sees all his comments addressed. The modified manuscript has significantly improved with respect to level of scientific scrutiny and overall quality.

The experimental data presentation has significantly improved and so has the data interpretation. In total, experimental data, DFT calculations, and thorough discussion of the results yield a coherent study on the role of water dissociation in CO₂RR for this type of electrocatalysts.

Response: We are grateful to the reviewer for endorsing our discussion and recommendation for publication of in Nature Communications.

Reviewer #3 (Remarks to the Author):

The authors have clearly explained all the questions except for one.

The article cited by the author on the stability of BOC (Angew. Chem. Int. Ed. e202214959 (2022)) also describes the partial reduction of BOC to Bi/BOC binary compound under cathodic reaction conditions. The author mentioned that (1) the reduced Bi could be oxidized to Bi(OH)₃ due to the local pH effect and (2) the reduced Bi can be oxidized to BOC with the presence of CO₂. I also agree that these side reactions can occur during the CO₂ reduction reaction. However, if these reactions happen, the total faradaic efficiency should be lower than 100% and more significantly, these repeated redox reactions will result in the reconstruction of crystal structure and morphology. The single crystalline nature of BOC is very likely altered from its original form (not detectable by in-situ Raman or XRD measurements).

Since the key point of the manuscript, that water dissociation becomes the RDS of the CO₂ reduction reaction, is well explained, this manuscript in its current form can be accepted. However, for the sake of completeness, I strongly suggest confirming the change in the crystal structure after the reaction by STEM measurement.

Response: We thank the reviewer for this valuable comment and support on the publication of

this work. In light of the reviewer's suggestion, TEM and corresponding SAED measurements on BOC-2 after stability test were implemented to monitor the crystal structure state. As shown in Figure R1, BOC-2 exhibits clear lattice fringes and single crystalline property while lattice disorder induced by V_O can be observed as well, which is quite similar to BOC-2 before CO_2RR (Figure R2). This indiscernible variation in SEM and TEM indicates that there is no obvious change in morphology and crystal structure in BOC-2 after stability test.

Moreover, the surface state of BOC-2 after evaluation was also studied. As can be seen from Figure R3a, the chemical state of Bi for BOC-2 after CO_2RR can keep well, indicating the stability on the surface of BOC-2 with V_O . This result is consistent with our in-situ Raman experiment (Figure R3b). Overall, the BOC-2 shows good stability during CO_2RR without obvious change in morphology, crystal structure and chemical state. These supplementary characterization results have been added in the revised Supplementary Information.

On the other hand, even though the repeated redox reaction of Bi may happen during CO_2RR , the undetectable change in XRD, SEM, TEM, XPS and Raman characterizations suggests that this redox is a fast and reversible process, which manifests the good stability of BOC-2. While the possible variation is hard to be observed from ex-situ characterizations. Therefore, more in-situ time-resolved characterizations need to be developed in this field, and we will make more efforts to study the mechanism of stability in the future.

Figure R1. a SEM image of BOC-2 after CO_2RR . b HRTEM image of BOC-2 after CO_2RR and the inset is the corresponding selected area electron diffraction (SAED) (corresponding to Supplementary Fig. 16 in the revised Supplementary Information).

Figure R2. a SEM image of BOC-2 before CO₂RR. b TEM image of BOC-2 before CO₂RR reaction and the inset is the corresponding selected area electron diffraction.

Figure R3. a XPS spectra of BOC-2 before and after CO₂RR. b in-situ Raman experiments of BOC-2 during CO₂RR (corresponding to Supplementary Fig. 16 in the revised Supplementary Information).